# Allelic strengths of encephalopathy-associated *UBA5* variants correlate between in vivo and in vitro assays

Xueyang Pan[1,2], Albert N Alvarez[3], Mengqi Ma[1,2], Shenzhao Lu[1,2], Michael W Crawford[3], Lauren C Briere[4], Oguz Kanca[1,2], Shinya Yamamoto[1,2,5], David A Sweetser[4,6], Jenny L Wilson[7], Ruth J Napier[3,8,9], Jonathan N Pruneda[3]*, Hugo J Bellen[1,2,5]*

[1]Department of Molecular and Human Genetics, Baylor College of Medicine, Houston, United States; [2]Jan & Dan Duncan Neurological Research Institute, Texas Children's Hospital, Houston, United States; [3]Department of Molecular Microbiology & Immunology, Oregon Health & Science University, Portland, United States; [4]Center for Genomic Medicine, Massachusetts General Hospital, Boston, United States; [5]Department of Neuroscience, Baylor College of Medicine, Houston, United States; [6]Division of Medical Genetics & Metabolism, Massachusetts General Hospital for Children, Boston, United States; [7]Division of Pediatric Neurology, Department of Pediatrics, Oregon Health & Science University, Portland, United States; [8]VA Portland Health Care System, Portland, United States; [9]Division of Arthritis & Rheumatic Diseases, Oregon Health & Science University, Portland, United States

*For correspondence:
pruneda@ohsu.edu (JNP);
hbellen@bcm.edu (HJB)

**Abstract** Protein UFMylation downstream of the E1 enzyme UBA5 plays essential roles in development and endoplasmic reticulum stress. Variants in the *UBA5* gene are associated with developmental and epileptic encephalopathy 44 (DEE44), an autosomal recessive disorder characterized by early-onset encephalopathy, movement abnormalities, global developmental delay, intellectual disability, and seizures. DEE44 is caused by at least 12 different missense variants described as loss of function (LoF), but the relationships between genotypes and molecular or clinical phenotypes remain to be established. We developed a humanized *UBA5* fly model and biochemical activity assays in order to describe in vivo and in vitro genotype–phenotype relationships across the *UBA5* allelic series. In vivo, we observed a broad spectrum of phenotypes in viability, developmental timing, lifespan, locomotor activity, and bang sensitivity. A range of functional effects was also observed in vitro across comprehensive biochemical assays for protein stability, ATP binding, UFM1 activation, and UFM1 transthiolation. Importantly, there is a strong correlation between in vivo and in vitro phenotypes, establishing a classification of LoF variants into mild, intermediate, and severe allelic strengths. By systemically evaluating *UBA5* variants across in vivo and in vitro platforms, this study provides a foundation for more basic and translational UBA5 research, as well as a basis for evaluating current and future individuals afflicted with this rare disease.

## eLife assessment

The authors establish a *Drosophila* model to assess the severity of disease-linked alleles of Uba5. Using both in vivo and in vitro experiments, this **valuable** study demonstrates the alleles fall into mild, intermediate, and severe classes, with **convincing** evidence to support their conclusion. This well-executed study establishes a model for further characterization of Uba5-related phenotypes in a powerful model system.

**eLife digest** Although rare diseases only impact a small fraction of the population, they still affect hundreds of millions of people around the world. Many of these conditions are caused by variations in inherited genetic material, which nowadays can be readily detected using advanced sequencing technologies. However, establishing a connection between these genetic changes and the disease they cause often requires further in-depth study.

One such rare inherited disorder is developmental and epileptic encephalopathy type 44 (DEE44), which is caused by genetic variations within the gene for UBA5 (short for ubiquitin-like modifier activating enzyme 5). For DEE44 to occur, both copies of the gene for UBA5, known as alleles, must contain one or more detrimental variation. Although all these variations prevent UBA5 from working correctly, the level of disruption they cause, known as allelic strength, varies between them. However, it remained unclear whether the severity of the DEE44 disease directly corresponds with the allelic strength of these variants.

To answer this question, Pan et al. tested how different genetic variants found in patients with DEE44 affected the behavior and health of fruit flies. These results were then compared against in vitro biochemical assays testing how alleles containing these variants impacted the function of UBA5.

When the fly gene for the enzyme was replaced with the human gene containing variations associated with DEE44, flies exhibited changes in their survival rates, developmental progress, lifespan, and neurological well-being. However, not all of the variants caused ill effects. Using this information, the patient variants were classified into three categories based on the severity of their effect: mild, intermediate, and severe. Biochemical assays supported this classification and revealed that the variants that caused more severe symptoms tended to inhibit the activity of UBA5 more significantly.

Pan et al. further analyzed the nature of the variants in the patients and showed that most patients typically carried one mild and one strong variant, although some individuals did have two intermediate variants. Notably, no patients carried two severe variants. This indicates that DEE44 is the result of UBA5 only partially losing its ability to work correctly.

The study by Pan et al. provides a framework for assessing the impact of genetic variants associated with DEE44, aiding the diagnosis and treatment of the disorder. However, further research involving more patients, more detailed clinical data, and testing other newly identified DEE44-causing variants is needed to solidify the correlation between allelic strength and disease severity.

## Introduction

Variants in the human *ubiquitin like modifier activating enzyme 5* (*UBA5*) gene have been associated with three autosomal recessive disorders. In most reported cases, biallelic *UBA5* variants cause developmental and epileptic encephalopathy 44 (DEE44, OMIM: #617132). The disease is characterized by early-onset encephalopathy, movement abnormalities, global developmental delay, and intellectual disability. Many individuals also have seizures, failure to thrive, and microcephaly. Delayed myelination, thinning of the corpus callosum, and white matter hyperintensities have also been documented with magnetic resonance imaging (MRI) (*Colin et al., 2016*; *Muona et al., 2016*). Biallelic *UBA5* has also been associated with spinocerebellar ataxia 24 (OMIM: #617133), which is characterized by a childhood-onset gait and limb ataxia (*Duan et al., 2016*). Another family has been reported with a rare homozygous missense variant in *UBA5* that segregates with severe congenital neuropathy (*Cabrera-Serrano et al., 2020*).

UBA5 is a key component in UFMylation, a post-translational modification pathway mediated by a ubiquitin-like protein (UBL) ubiquitin fold modifier 1 (UFM1) (*Millrine et al., 2023*). UBL modifications play an essential role in eukaryotic biology by regulating protein stability and function via different enzymatic complexes (*Cappadocia and Lima, 2018*; *van der Veen and Ploegh, 2012*). UFMylation is conserved in metazoans and plants (*Figure 1A*; *Millrine et al., 2023*). In this pathway, UFM1 is first proteolytically processed by a UFM1-specific peptidase 1/2 (UFSP1/2) to expose a C-terminal Gly (*Kang et al., 2007*; *Komatsu et al., 2004*; *Millrine et al., 2022*). The ensuing conjugation process involves three steps. The first two steps are facilitated by UBA5, an E1 activating enzyme specific to UFMylation. UBA5 activates UFM1 through ATP-dependent adenylation of the UFM1 C-terminal Gly, which is then transferred onto the UBA5 active site Cys 250, forming a high-energy thioester

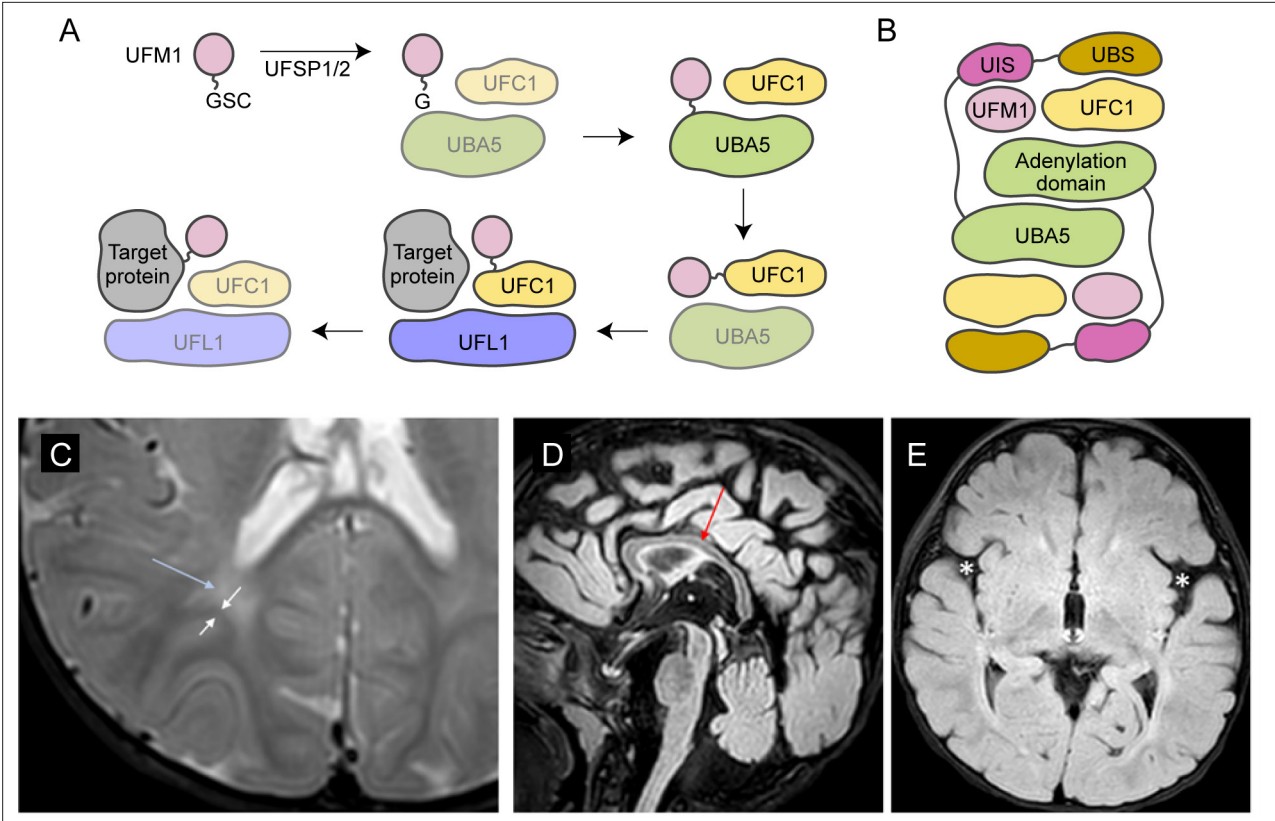

**Figure 1.** Brain magnetic resonance imaging (MRI) images of the proband. (**A**) A diagram showing the UFMylation pathway. Details of the biochemical processes in the pathway are described in the main text. In UBA5 proteins, only the adenylation domains are shown in the diagram. (**B**) A diagram of the UBA5:UFM1:UFC1 complex. In the complex, two copies of UBA5 form a homodimer that interacts with ubiquitin fold modifier 1 (UFM1) via a trans-binding mechanism. The activation of UFM1 requires the adenylation domain of one UBA5 subunit and the UFM1-interacting sequence (UIS) of the other UBA5 subunit in the complex. The opposing protomer of the UBA5 homodimer also contributes a UFC1-binding sequence (UBS) that is required for UFM1 transthiolation. (**C**) Axial T2 image showing periventricular T2 hyperintensity (blue arrow) resulting in prominence of the subcortical U-fibers (white arrows). (**D**) Sagittal flair showing mild thinning of the corpus callosum (red arrow). (**E**) Axial flair image demonstrating widening of the sylvian fissures (white asterisks).

intermediate. Next, the UFM1-specific E2 conjugating enzyme, UFC1, binds to the activated UBA5– UFM1 intermediate and receives UFM1 onto its active site Cys through a transthiolation reaction (*Figure 1A*; *Gavin et al., 2014*; *Komatsu et al., 2004*). The UFM1 activation and transthiolation processes are achieved by a trans-binding mechanism involving two molecules each of UBA5, UFM1, and UFC1, wherein one UBA5 protomer performs the enzymatic processes while the other provides essential UFM1- and UFC1-binding sites in trans (*Figure 1B*; *Kumar et al., 2021*; *Mashahreh et al., 2018*; *Oweis et al., 2016*). Next, the E3 ligase UFL1 functions as a scaffold to bring the activated UFC1–UFM1 conjugate to the substrate protein and facilitate the conjugation of UFM1 to a substrate Lys residue (*Peter et al., 2022*; *Tatsumi et al., 2010*).

UFMylation has been implicated in regulating many processes such as genome stability and receptor activation (*Liu et al., 2020*; *Qin et al., 2019*; *Wang et al., 2019*; *Yoo et al., 2014*), but the principal role is believed to be in regulating proteotoxic stress at the endoplasmic reticulum (ER), where UFMylation of stalled ribosomes initiates quality control measures (*Liang et al., 2020*; *Scavone et al., 2023*; *Walczak et al., 2019*; *Wang et al., 2020*). So far, the only well-characterized *Uba5* mutant animal model is a mouse model in which removal of the gene causes embryonic lethality due to hematopoietic defects (*Tatsumi et al., 2011*). Tissue-specific mouse knockouts of other genes within the UFMylation pathway support a role in regulating ER stress within secretory cells, as well as a critical developmental role within the central nervous system (CNS) (*Muona et al., 2016*; *Zhu et al., 2019*).

**Table 1.** Summary of genotypes of the reported cases.

| References | Family | Allele #1 | Allele #2 |
|---|---|---|---|
| *Colin et al., 2016* | A | p.Ala371Thr (IA*) | p.Gln302* |
| | B | p.Ala371Thr (IA) | p.Lys324Asnfs*14 |
| | C | p.Asp389Tyr (IA) | p.Val260Met (II) |
| | D | p.Met57Val (II) | p.Gly168Glu (III) |
| *Muona et al., 2016* | A | p.Ala371Thr (IA) | p.Arg55His (III) |
| | B | p.Ala371Thr (IA) | p.Tyr285* |
| | C, E | p.Ala371Thr (IA) | p.Arg188* |
| | D | p.Ala371Thr (IA) | p.Arg61* |
| *Arnadottir et al., 2017* | | p.Ala371Thr (IA) | p.Ala288 = (splicing variant) |
| *Daida et al., 2018* | | p.Tyr72Cys (IB) | Deletion |
| *Mignon-Ravix et al., 2018* | | p.Tyr53Phe (II)[†] | p.Tyr53Phe (II) |
| *Low et al., 2019* | | p.Asp389Gly (IA) | Deletion |
| *Briere et al., 2021* | A | p.Ala371Thr (IA) | p.Cys303Arg (III) |
| | B | p.Ala371Thr (IA) | p.Arg188* |
| | C | p.Ala371Thr (IA) | p.Leu254Pro (III) |
| | D | p.Ala371Thr (IA) | p.Cys303Arg (III) |
| This study | | p.Met57Val (II) | p.Gln312Leu (IB) |

*The variant classification using fly phenotypic assays (results shown in **Figure 4**).
[†]Consanguineous family.

To date, 24 individuals from 17 families who have *UBA5*-associated DEE44 have been reported (*Arnadottir et al., 2017*; *Briere et al., 2021*; *Colin et al., 2016*; *Daida et al., 2018*; *Low et al., 2019*; *Mignon-Ravix et al., 2018*; *Muona et al., 2016*). The genotypes and clinical features of the affected individuals are summarized in *Table 1* and *Supplementary file 1*, respectively. Prior functional studies using cultured cells or patient cells show that many reported *UBA5* variants cause various levels of loss of function (LoF). However, the study of the genotype–phenotype relationship is hampered by the limited number of affected individuals, incomplete description of clinical presentations and the heterogeneous genetic background. Variant-specific in vivo models are powerful tools for studying genotype–phenotype relationship, especially for rare diseases (*Arnadottir et al., 2017*; *Goodman et al., 2021*; *Lu et al., 2022a*; *Lu et al., 2022b*; *Ma et al., 2023*; *Tepe et al., 2023*). However, systematic assessment of the effects of disease-causing variants in vivo is a challenge as it can be very labor intensive. Moreover, the in vivo assays should be compared to functional studies of the variant proteins, which typically relies on biochemical or other cell-based assays that are not available for most proteins/genes. By combining phenotypic studies and biochemical assays it should be possible to assess the severity of each variant, providing valuable information for the affected individuals and for assessing possible therapeutic interventions. In addition, genotype–phenotype relationships offer information about the molecular basis underlying variants LoF, paving the way for future therapeutic development.

In this study, we assess the genotype–phenotype relationship in *UBA5*-associated DEE44 variants by determining the phenotypes of variant-specific fruit fly models. In conjunction with the in vivo data, we also comprehensively assess the biochemical properties of each variant using assays that report on protein stability, ATP binding, UFM1 activation, and UFM1 transthiolation. The presence and severity of the phenotypes in flies are highly variant dependent. Similarly, the enzymatic activities of the variants vary widely in vitro. Interestingly, both in vivo and in vitro assays produce a very similar allelic series for the variants, suggesting a correlation between specific enzymatic properties and the phenotypes in the animal models. Finally, combining our animal model work with available insights

**Table 2.** Bioinformatic predictions of the pathogenicity of reported *UBA5* variants.

|  | Variant 1 | Variant 2 |
|---|---|---|
| Genomic position (GRCh38) | 3:132665830A>G | 3:132394214A>T |
| Amino acid change | p.Met57Val | p.Gln312Leu |
| Allele frequency in gnomAD | Absent | Absent |
| CADD score | 25.3 | 29.1 |
| SIFT | Damaging | Damaging |
| PolyPhen2 | 1.000 (probably damaging) | 0.999 (probably damaging) |
| MutationTaster | Disease causing | Disease causing |
| PROVEAN | −3.18 (deleterious) | −6.59 (deleterious) |

into UBA5 enzymology provides us with a much better understanding of the structure–function relationship of the UBA5 variants.

## Results

### Report of an individual with compound heterozygous *UBA5* missense variants

The reported proband is a 3-year boy with axial hypotonia, generalized dystonia and lower extremity spasticity, global developmental delay, esotropia, and failure to thrive. In infancy he had frequent vomiting with feeds and failure to thrive requiring gastrostomy tube placement. At 4 months of age developmental delay and abnormal tone were noted. He has continued to have difficulty with weight gain and growth, although he is not microcephalic. At 30 months of age, he rolls occasionally but does not sit, has a limited ability to grab objects, babbles, and follows familiar commands. There has been no regression. He has axial hypotonia, generalized dystonia, and lower extremity spasticity. An electroencephalogram at 7 months of age showed multifocal epileptiform discharges in drowsiness and sleep. He has not had seizures. A trial of carbidopa–levodopa was initially thought to be helpful, but ultimately tapered off without worsening. He is taking baclofen at night which helps with sleep. His MRI studies at 7 and 23 months of age were read as normal, although on review show a slightly thin corpus callosum, posterior periventricular white matter T2 hyperintensity resulting in increased conspicuity of the subcortical U-fibers and widening of the Sylvian fissures (*Figure 1C–E*).

On trio exome sequencing, the proband was found to have compound heterozygous variants in UBA5, NM_024818.6:c.169A>G (p.Met57Val) and c.935A>T (p.Gln312Leu). The p.Met57Val variant was inherited from his mother and the p.Gln312Leu variant was inherited from his father. The p.Met57Val variant has been reported in one individual with DEE44 (*Colin et al., 2016*). The p.Gln312Leu variant has not been previously reported. Neither variant is reported in the gnomAD database v.2.1.1 (https://gnomad.broadinstitute.org). Both variants are predicted to be damaging or probably damaging by multiple pathogenicity prediction tools (*Table 2*). In addition, chromosomal microarray revealed a del(4)(q13.2q13.3) copy number variant, which was inherited from his healthy mother.

### Establishment of a variant-specific *UBA5*-associated disease model in fruit flies

To investigate the functions of *UBA5* variants in vivo, we utilized *Drosophila melanogaster* as a model organism. *Uba5* is the ortholog of human *UBA5* in flies (*UBA5* refers to the human gene; *Uba5* refers to the fly gene). The two proteins share 64% identity and 75% similarity in amino acid sequence, and the *Drosophila* Integrative Ortholog Prediction Tool (DIOPT) score between *UBA5* and *Uba5* is 15/16, indicating a high degree of homology (*Figure 2A*; *Hu et al., 2021*). The UBA5 protein has an adenylation domain, a UFM1-interacting sequence (UIS) and a UFC1-binding sequence (UBS), all of which are required for UFM1 activation and transthiolation (*Bacik et al., 2010*; *Habisov et al., 2016*; *Kumar et al., 2021*; *Padala et al., 2017*; *Xie, 2014*; *Figure 2A*). Similarly, Uba5 has all three highly

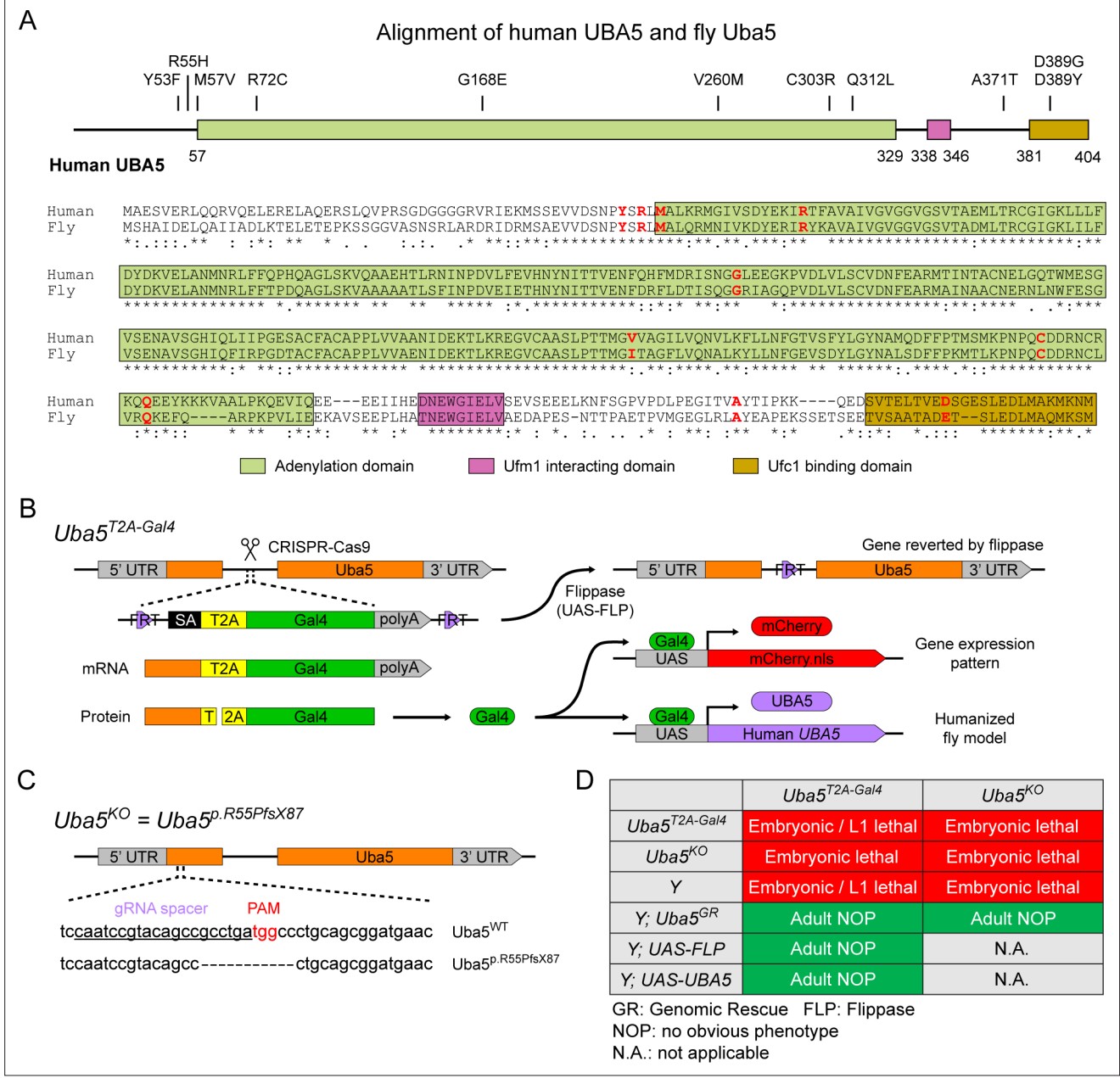

**Figure 2.** UFMylation pathway, conservation of UBA5, and generation of fly *Uba5* loss of function (LoF) alleles. (**A**) Alignment of the human UBA5 and fly Uba5 protein sequences. The functional domains of UBA5 are marked in colored boxes. The developmental and epileptic encephalopathy 44 (DEE44)-associated variants are marked in the protein topology diagram and the protein sequence alignment (letters in red). (**B**) Generation of the *Uba5^(T2A-Gal4)* allele and the uses of the allele in flippase (FLP)-mediated conversion. The expression of the GAL4 to drive a fluorescent protein allows assessment of gene expression, and humanization of the flies by expression of human *UBA5* cDNA. (**C**) Generation of *Uba5* null allele by CRISPR-mediated indel formation. (**D**) Loss of *Uba5* causes lethality in early developmental stage. The lethality is rescued by a genomic rescue construct, the expression of FLP (*Uba5^(T2A-Gal4)* mutants only), and the expression of human *UBA5* cDNA.

conserved functional domains, and all of the amino acid residues affected by the DEE44-associated variants reported so far are conserved in the fly protein (*Figure 2A*).

To study the variant-specific functions, we generated humanized fruit fly models in which the expression of the endogenous *Uba5* gene is removed or severely suppressed and a human *UBA5* cDNA is expressed under the control of the endogenous *Uba5* enhancer and promoter. If the human reference UBA5 functions in flies and rescues the *Uba5* severe LoF phenotypes, the DEE44-associated variants can be expressed and their functions can be assessed by the phenotypes of flies. To achieve this, we

generated a *Uba5^T2A-Gal4* allele using a CRISPR-Mediated Integration Cassette (CRIMIC) strategy (*Lee et al., 2018*). In the *Uba5^T2A-Gal4* allele, an *FRT-Splice Acceptor (SA)-T2A-GAL4-polyA-FRT* cassette was inserted into a coding intron of the *Uba5* gene. The SA causes the inclusion of the cassette during transcription, while the polyA sequence arrests the transcription generating a truncated transcript. The translation of the transcript is arrested at the viral ribosomal skipping site (T2A) and reinitiated after the site, producing an untagged GAL4 protein (*Figure 2B*; *Diao et al., 2015*; *Lee et al., 2018*). The *Uba5^T2A-Gal4* allele is likely a severe LoF allele (see below) (*Lee et al., 2018*). In addition, this allele also results in the expression of GAL4 under the control of the endogenous *Uba5* enhancer and promoter, which enables the assessment of native gene expression pattern as well as the expression of human *UBA5* cDNA (*Figure 2B*). We also generated a *Uba5* null allele by CRISPR-induced indel formation (*Uba5^p.Arg55Profs*87*, named *Uba5^KO*) (*Figure 2C*).

We first tested the viability of the flies with the *Uba5^T2A-Gal4* and the *Uba5^KO* alleles. The fly *Uba5* gene is located on the X chromosome. For both alleles, homozygous female and hemizygous male flies are lethal at the embryonic stage, although a few *Uba5^T2A-Gal4* escapers survive to the L1 larval stage (*Figure 2D*). The lethality is rescued by a genomic rescue (GR) construct that carries the *Uba5* locus (P[acman] clone CH321-02B13) (*Venken et al., 2010*), indicating that the lethality in both lines is caused by the LoF of *Uba5*. Moreover, expression of flippase (FLP) using *Uba5^T2A-Gal4* removes the insertion of the CRIMIC cassette and reverts the lethality of the *Uba5^T2A-Gal4* hemizygous males, showing that the lethality is indeed caused by the *Uba5^T2A-Gal4* allele (*Figure 2D*; *Lee et al., 2018*). Finally, expression of reference human *UBA5* cDNA using *Uba5^T2A-Gal4* rescues the lethality of the mutants, showing that the functions of the fly and human proteins are evolutionarily conserved (*Figure 2D*).

## *Uba5* is expressed in a subset of neurons and glia in the fly CNS

Next, we examined the expression pattern of *Uba5* by expressing a nuclear localized mCherry fluorescent protein (*UAS-mCherry.nls*) under the control of *Uba5^T2A-Gal4*. *Uba5* is expressed in multiple tissues in L3 larvae and adult flies (*Figure 3A*), consistent with high-throughput gene expression profiling results (*Figure 3—figure supplement 1A*; *Leader et al., 2018*; *Li et al., 2022*). We next analyzed the expression of *Uba5* in the CNS. We stained the *Uba5^T2A-Gal4*>mCherry.nls larval CNS and adult brain with anti-Elav and anti-Repo antibodies to mark the nuclei of neurons and glial cells, respectively. In both larval CNS and adult brain, the mCherry.nls signals are found in a subset of neurons and glia (*Figure 3B, C*), suggesting that *Uba5* is expressed in the fly CNS but not in all cells. This finding is also consistent with the previously published single-cell RNA sequencing profile (*Figure 3—figure supplement 1B*; *Davie et al., 2018*). *Uba5* is expressed more widely in the adult brains than in the larval CNS. In larval CNS, it is expressed in many fewer neurons than Elav (*Figure 3A, B*). The expression pattern resembles that of the *para* gene, which encodes the sole voltage-gated sodium channel in *Drosophila* that is only expressed in differentiated, actively firing neurons (*Ravenscroft et al., 2020*). This suggests that UBA5 may be required for the activity of neurons.

Since *Uba5* is expressed in multiple tissues in flies, we sought to determine the tissue-specific requirement of *Uba5* for fly development. We expressed the reference *UBA5* cDNA in *Uba5^KO* hemizygous mutants using various GAL4 drivers and examined the viability of the animals. Ubiquitous expression of *UBA5* by *da-Gal4* or *Act-Gal4* fully rescues the lethality of the mutant animals. However, tissue-specific expression of *UBA5* in fat body (*lpp-Gal4*), muscles (*Mef2-Gal4*), neurons (*Elav-Gal4*), or glial cells (*Repo-Gal4*) fails to rescue the lethality (*Figure 3—figure supplement 1C*). Hence, we conclude that *Uba5* is required in multiple tissues. Interestingly, overexpression of *UBA5* in *Uba5^KO/+* heterozygous flies does not cause any obvious phenotype (*Figure 3—figure supplement 1C*), showing that *Uba5* overexpression is not overtly toxic.

## DEE44-associated variants exhibit different rescuing abilities in flies

Next, we sought to evaluate the function of *UBA5* variants using the humanized fly model (*Bellen and Yamamoto, 2015*). We expressed reference or variant *UBA5* cDNA and DEE44-associated *UBA5* variants using *Uba5^T2A-Gal4* and measured phenotypes including survival rate, developmental timing, lifespan, locomotor activity, and seizure-like activity following mechanical stimulation in *Uba5^T2A-Gal4* hemizygous male flies. The variants we tested include all previously reported variants (*Arnadottir et al., 2017*; *Briere et al., 2021*; *Colin et al., 2016*; *Daida et al., 2018*; *Low et al., 2019*; *Mignon-Ravix*

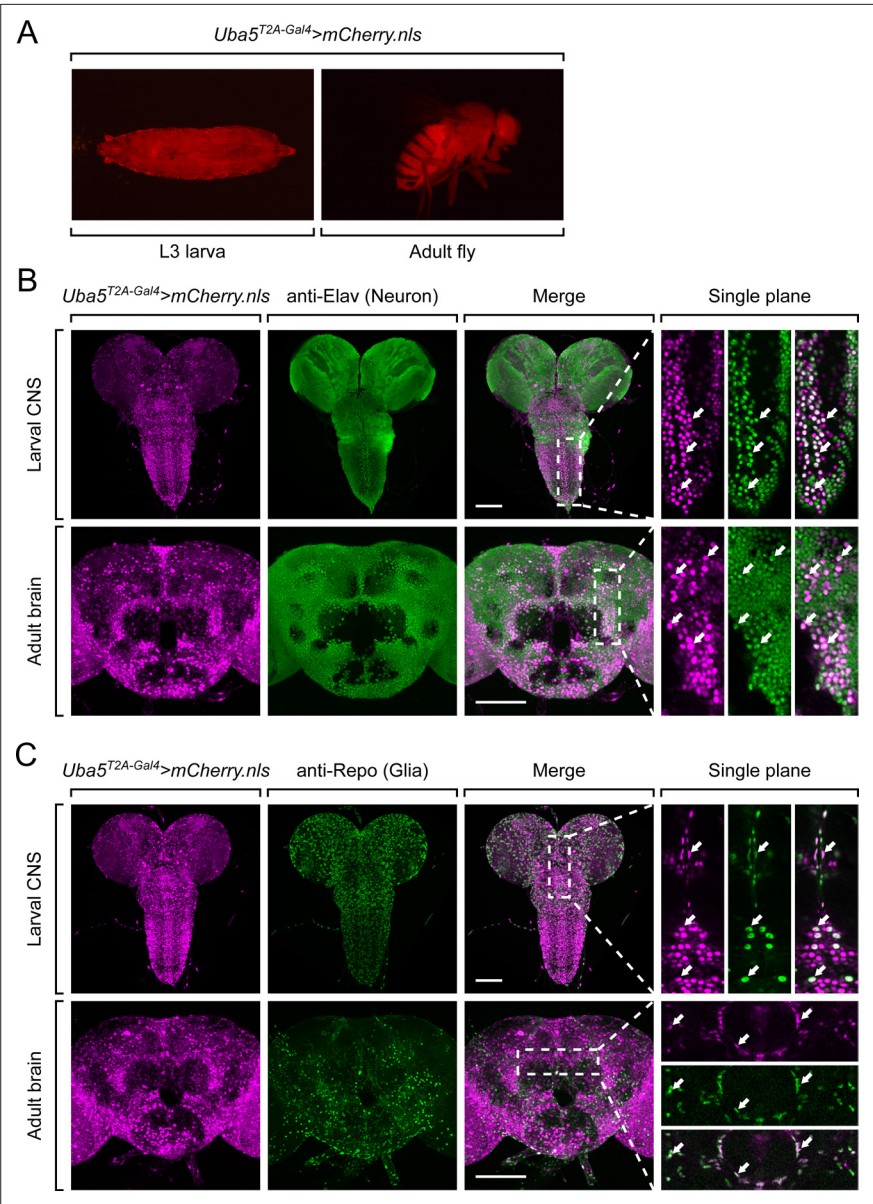

**Figure 3.** *Uba5* is expressed in a subset of neurons and glial cells in fly central nervous system (CNS). (**A**) The expression of nuclear localized mCherry (mCherry.nls) driven by the *Uba5^{T2A-Gal4}* allele (*Uba5^{T2A-Gal4}>mCherry. nls*) shows that *Uba5* is expressed in L3 larvae and adult flies. (**B, C**) The larval CNS and adult brain of *Uba5^{T2A-Gal4}>mCherry.nls* animals were immunostained with a neuronal (Elav, Panel B) or glial marker (Repo, Panel C). Maximum projections of confocal z-stack images are shown. Single plane, high magnification images of the regions indicated by the dashed squares are shown on the right to visualize the colocalizations between mCherry and the immunostaining signals. Arrows indicate cells that colocalize both markers. Scale bar, 100 μm.

The online version of this article includes the following figure supplement(s) for figure 3:

**Figure supplement 1.** Single-cell gene expression pattern of *Uba5* and the rescue of *Uba5* mutants by tissue-specific *UBA5* expression.

et al., 2018; *Muona et al., 2016*), as well as a novel variant from the individual we report in this study (p.Gln312Leu).

We first assessed the ability of the variants to rescue the lethality of *Uba5^{T2A-Gal4}* mutants. Flies were raised at three different temperatures: 29, 25, and 18°C. Lower temperatures lead to lower expression levels of *UBA5* when compared to higher temperatures (*Nagarkar-Jaiswal et al., 2015*). At all temperatures, a synthetic enzyme-dead *UBA5* mutant p.Cys250Ala failed to rescue lethality

(*Figure 4A*). Similarly, four DEE44-associated variants (p.Arg55His, p.Gly168Glu, p.Leu254Pro, and p.Cys303Arg) failed to rescue the lethality at all temperatures tested, indicating that they are severe LoF. Three variants (p.Tyr53Phe, p.Met57Val, and p.Val260Met) partially rescued the lethality at 18°C and the survival rate increased at elevated temperatures (*Figure 4A*). This suggests that the three variants are likely hypomorphic alleles and supports a dosage-sensitive effect of *Uba5*. However, six variants (p.Arg72Cys, p.Val260Met, p.Gln312Leu, p.Ala371Thr, p.Asp389Gly, and p.Asp389Tyr) fully rescued lethality, suggesting they are mild LoF or do not affect protein function. Based on these results, we stratified the variants into three groups according to the activity of rescuing lethality: Group I, full rescue; Group II, partial rescue; Group III, failure to rescue (*Table 3*, Group I was further divided into IA and IB according to the other phenotypes described below).

The variants that survive to adults were next tested for the time it takes for animals to eclose as adults, and lifespan. The variants in Group II exhibited significant developmental delay as well as a shortened lifespan, showing that they are partial LoF (*Figure 4B, C*). In contrast, some variants in Group I (Group 1A: p.Ala371Thr, p.Asp389Gly, and p.Asp389Tyr) caused neither defect. Other Group I variants (Group 1B: p.Arg72Cys and p.Gln312Leu) caused a shortened lifespan but did not affect the timing of development (*Figure 4B, C*), indicating that they are also partial LoF variants but could cause milder defects than Group II variants.

To determine if the flies display features that are associated with dysfunction of the nervous system, we measured locomotor activity using a climbing assay and assessed susceptibility to seizures using a bang sensitivity assay (*Song and Tanouye, 2008*). Flies with variants in Groups IB and II displayed reduced climbing activity at Day 7 and more severe defects by Day 30, showing a progressive worsening of the defects (*Figure 4D*). Moreover, the variants in Group II exhibited a bang-sensitive phenotype by displaying seizure-like behavior and paralysis following mechanical stimulation (*Figure 4E*). However, the Group IA variants displayed neither a climbing defect nor bang sensitivity (*Figure 4D, E*). These results are consistent with our classification of variants based on other assays: Group IA, no obvious LoF or benign; Groups IB and II, intermediate LoF; and Group III, severe LoF.

Given the observed neurological defects, we tested whether the variants affect the synapse morphology of the neuromuscular junctions (NMJs) in third instar larvae. Although a previous study reported that knockdown of *Uba5* in motor neurons caused decreased number and increased size of synaptic boutons at the NMJs (*Duan et al., 2016*), we did not observe any obvious changes in bouton number or size caused by the Group II variants (*Figure 4—figure supplement 1A–C*). Hence, we conclude that the synaptic growth of fly larvae is not affected by the *UBA5* variants.

## Structural analysis of UBA5 variants

In order to link our findings in the fly model with functional changes in UBA5, we first analyzed the potential structural changes caused by the UBA5 variants. Extensive structural analyses have been performed on UBA5, including its ability to bind ATP and homodimerize within the adenylation domain, its interaction with UFM1 in the process of activation, and its engagement of UFC1 prior to UFM1 transthiolation (*Bacik et al., 2010*; *Kumar et al., 2021*; *Oweis et al., 2016*; *Padala et al., 2017*; *Soudah et al., 2019*; *Wesch et al., 2021*). This structural detail of UBA5 function offers a unique opportunity to visualize the location of variants and categorize their predicted effects. We compiled a series of UBA5 structures to create a composite model that illustrates (a) UBA5 homodimerization, (b) ATP coordination, (c) UFM1 binding, and (d) UFC1 binding (*Figure 5A*). Focusing on the UFM1 C-terminal Gly and the active site Cys residues of UBA5 and UFC1, the movements that occur during UFM1 activation and transthiolation can be mapped (*Figure 5A*, yellow spheres).

With this structural model as a basis, we highlighted the location of all UBA5 variants (*Figure 5A*, red spheres). Variants at positions Asp389 and Ala371 lie outside of the regions with determined structure, so their locations are modeled based on all available data. All Group IA variants localize in proximity of the UBS in protein sequence. The p.Asp389Gly and p.Asp389Tyr variants affect Asp389 which is two amino acids upstream of the structurally resolved UBS. For p.Ala371Thr, although there is no structure of this region, previous biochemical and crosslinking data have demonstrated its proximity to the UFC1 active site (*Kumar et al., 2021*). Four variants, p.Tyr53Phe, p.Arg55His, p.Met57Val, and p.Arg72Cys, affect residues near the ATP-binding site (*Figure 5B*). The side chain of Arg55 makes direct contacts to the bound ATP, while Met57 and Tyr53 make secondary contacts behind this site, and Arg72 makes more distant tertiary contacts. Substitutions at these positions, therefore, may affect

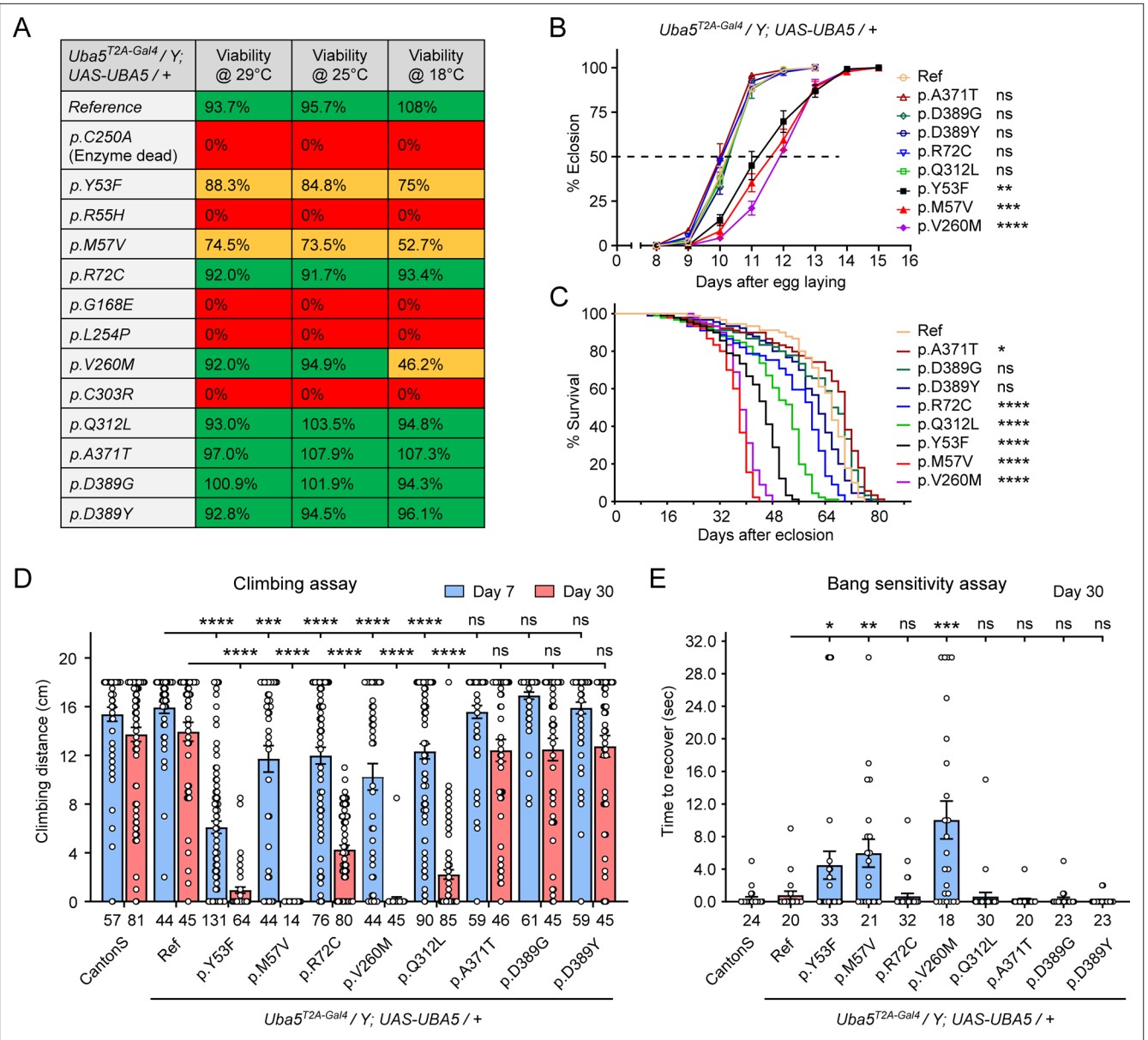

**Figure 4.** Developmental and epileptic encephalopathy 44 (DEE44)-associated variants exhibit different rescuing abilities in flies. (**A**) The DEE44-associated *UBA5* variants rescued the lethality of *Uba5* mutant flies with varying efficiency. *Uba5^{T2A-Gal4}/FM7* females were crossed with *UAS-UBA5* males and the viability of *Uba5^{T2A-Gal4}/Y; UAS-UBA5/+* progenies were measured by Mendelian ratio and indicated by color codes: red, zero viability; yellow, partial viability (<90% of expected number); green, full viability (90% and above). (**B**) Three variants caused developmental delay in *Uba5^{T2A-Gal4}/Y; UAS-UBA5/+* flies. The embryos were collected within 6 hr and the number of eclosed adult flies was counted at the same time every day. Three replicates were performed in each group. (**C**) Five variants caused reduced lifespan in *Uba5^{T2A-Gal4}/Y; UAS-UBA5/+* flies. (**D**) Five variants caused progressive climbing defects in *Uba5^{T2A-Gal4}/Y; UAS-UBA5/+* flies. Flies were tested on Days 7 and 30. The climbing activity of *CantonS* wildtype flies is shown as reference. Numbers of animals (*n* values) in each group are indicated under the bars. (**E**) Three variants caused a bang-sensitive phenotype in *Uba5^{T2A-Gal4}/Y; UAS-UBA5/+* flies. Flies were tested on Day 30. The bang sensitivity of *CantonS* wildtype flies is shown as reference. Numbers of animals (*n* values) in each group are indicated under the bars. (**B–E**) Flies were cultured under 25°C. The results of DEE4 variant-expressing flies are compared with the result of reference *UBA5*-expressing flies. Results are presented as means ± standard error of the mean (SEM). Statistical analyses were performed via two-sided, unpaired Student's *t*-test. ns, not significant; *p < 0.05; **p < 0.01; ***p < 0.001; ****p < 0.0001.

The online version of this article includes the following source data and figure supplement(s) for figure 4:

**Source data 1.** Source data for *Figure 4A–E*.

**Figure supplement 1.** The Group II *UBA5* variants do not cause obvious synaptic growth defects.

**Figure supplement 1—source data 1.** Source data for *Figure 4—figure supplement 1B, C*.

**Table 3.** Summary of phenotypes of humanized flies expressing *UBA5* variants.

|  | Variants | Survival rate | Dev. delay | Lifespan | Climbing defects | Bang sensitivity |
|---|---|---|---|---|---|---|
| Group IA | p.Ala371Thr | Normal | No | Normal | No | No |
|  | p.Asp389Gly | Normal | No | Normal | No | No |
|  | p.Asp389Tyr | Normal | No | Normal | No | No |
| Group IB | p.Arg72Cys | Normal | No | Decreased | Yes | No |
|  | p.Gln312Leu | Normal | No | Decreased | Yes | No |
| Group II | p.Tyr53Phe | Decreased | Yes | Decreased | Yes | Yes |
|  | p.Met57Val | Decreased | Yes | Decreased | Yes | Yes |
|  | p.Val260Met | Decreased | Yes | Decreased | Yes | Yes |
| Group III | p.Arg55His | Lethal | N.A. | N.A. | N.A. | N.A. |
|  | p.Gly168Glu | Lethal | N.A. | N.A. | N.A. | N.A. |
|  | p.Leu254Pro | Lethal | N.A. | N.A. | N.A. | N.A. |
|  | p.Cys303Arg | Lethal | N.A. | N.A. | N.A. | N.A. |

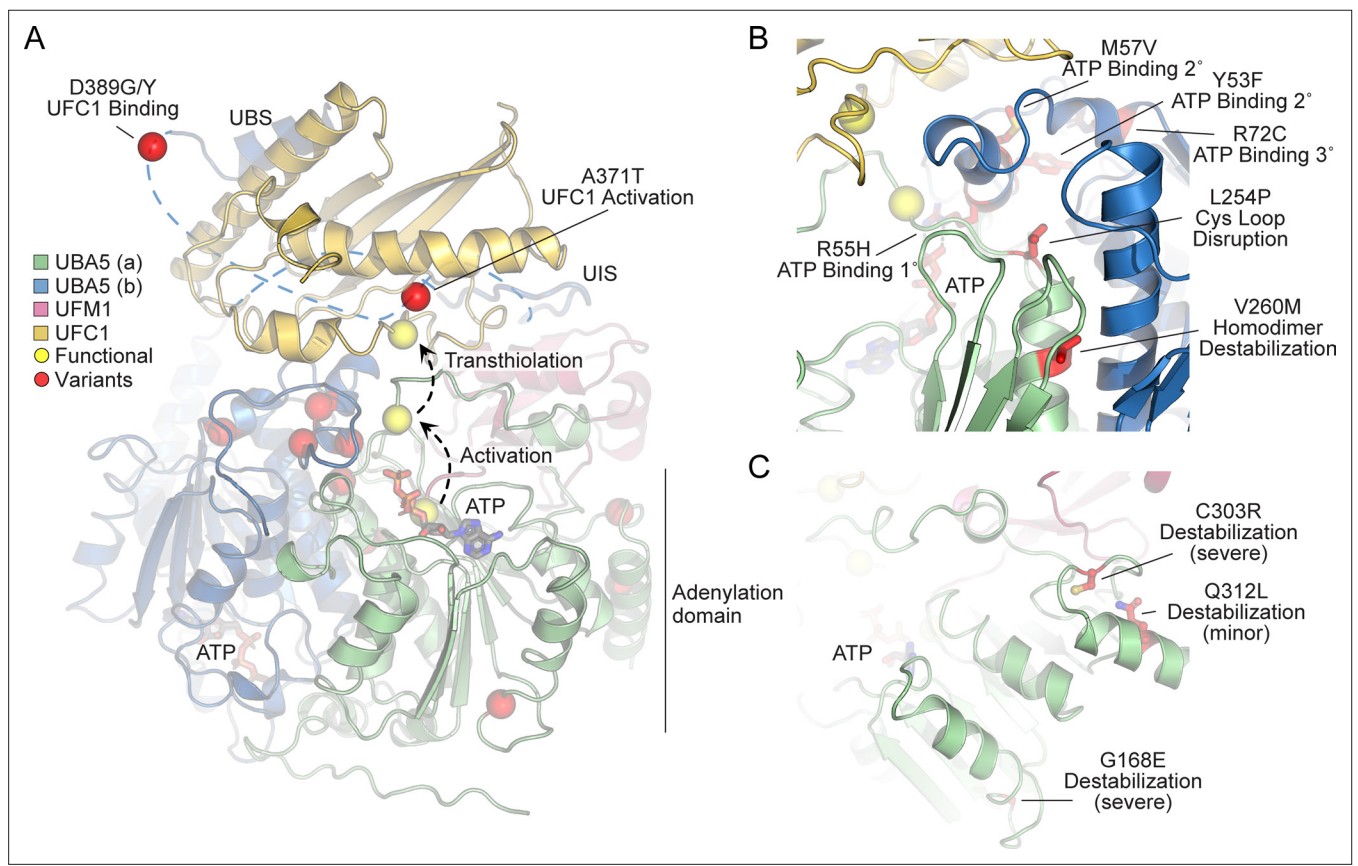

**Figure 5.** Structural analysis of UBA5 variants. (**A**) Composite model of a UBA5 homodimer (green and blue) bound to ATP (gray sticks), ubiquitin fold modifier 1 (UFM1; magenta), and UFC1 (gold). The model was built using a series of UBA5 complex structures with UFM1 and UFC1 (PDB 6H77, 7NW1, and a modeled UBA5:UFC1 complex; *Kumar et al., 2021*; *Soudah et al., 2019*). Functional residues comprising the active site cysteines of UBA5 and UFC1, as well as the C-terminus of UFM1 are shown in yellow spheres. UBA5 variants are shown in red spheres and are labeled with their predicted structural effects. (**B**) Close-up view of variants (red sticks) within the UBA5 active site (yellow sphere), ATP-binding pocket, and homodimerization interface. (**C**) Close-up view of variants (red sticks) expected to impact UBA5 protein stability (results shown in the following figures).

the affinity of UBA5 toward ATP. Residue Leu254 is four amino acids downstream of the active site Cys250, contained within a loop region that must undergo conformational changes to support UFM1 activation and subsequent transthiolation (*Figure 5A, B*). The Pro substitution within the p.Leu254Pro variant may constrain the required flexibility of the Cys250 loop, thus impacting ATP binding, UFM1 activation, and UFM1 transthiolation. Val260 is buried within the UBA5 homodimeric interface, opposing residue Val260 of the second protomer (*Figure 5B*). The increased size of the p.Val260Met variant may cause a steric clash that reduces UBA5 dimerization, which would decrease ATP binding, UFM1 activation, and UFM1 transthiolation as a result. Positions Cys303 and Gly168 are fully buried, making structural contacts to support the fold of the adenylation domain (*Figure 5C*). The increased size and charge associated with the p.Cys303Arg and p.Gly168Glu variants would most likely cause significant defects in UBA5 folding and stability. Lastly, the site of the novel p.Gln312Leu variant reported here is partially buried, making structural contacts to a loop region underneath the UFM1-binding site, and thus substitutions at this site may also cause structural instability (*Figure 5C*).

## Generation and characterization of purified UBA5 variant proteins

To determine the functional capacity of the UBA5 variants, we expressed and purified reference and variant UBA5 proteins from *Escherichia coli* for biochemical assays. Two severe LoF (Group III) variants p.Gly168Glu and p.Cys303Arg were insoluble in the protein purification process, indicating that the variants influence the stability and/or folding of these proteins. We were able to obtain pure and homogeneous samples of all other UBA5 constructs (*Figure 6A*). For our first measure of UBA5 function, we measured the effects of each variant on protein stability using a thermal shift assay. Under our assay conditions, reference UBA5 demonstrated a melting temperature (Tm) of 46°C. While most UBA5 variants exhibited little-to-no change in Tm, several showed a subtle destabilization by ≥1°C, including p.Arg72Cys, p.Leu254Pro, p.Val260Met, and p.Ala371Thr (*Figure 6B*). Interestingly, the Group IB variant p.Gln312Leu exhibited a minor unfolding at 36°C before fully melting at 48°C. This could indicate destabilization in the local structure surrounding the Gln312Leu substitution, while leaving the remaining protein structure intact.

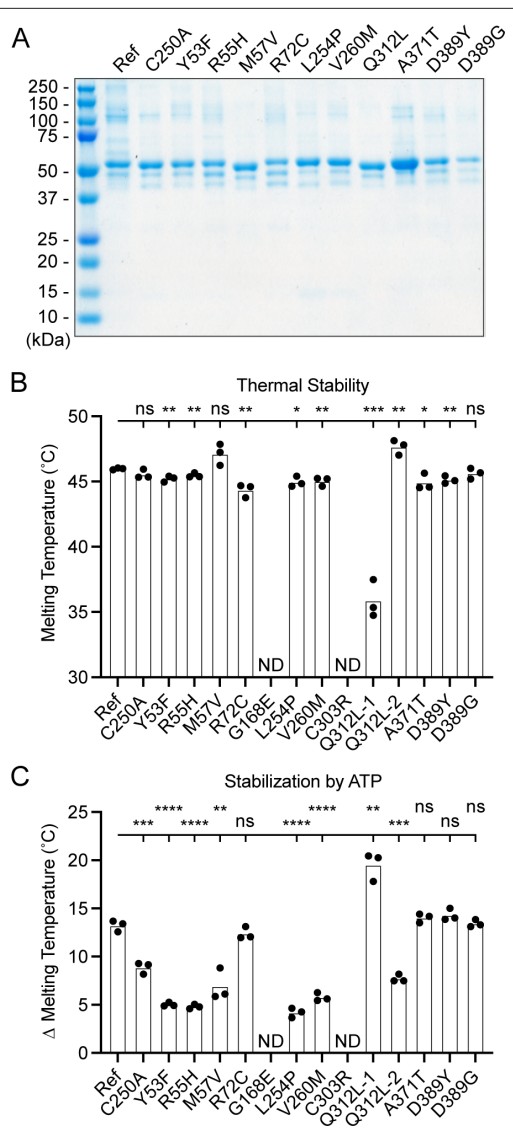

**Figure 6.** Preparation and stability of UBA5 variant proteins. (**A**) Coomassie-stained sodium dodecyl sulfate–polyacrylamide gel electrophoresis (SDS–PAGE) analysis of all purified UBA5 variant proteins. (**B**) Thermal shift assay measuring the melting temperature (Tm) of all UBA5 variant proteins, with the exception of p.Gly168Glu and p.Cys303Arg which could not be produced. The p.Gln312Leu variant displayed two melting curves. Experiments were performed in triplicate over three biological replicates. (**C**) Change in melting temperature for all UBA5 variants in the presence of 5 mM ATP. Upon ATP addition, the p.Gln312Leu variant transitioned to a single melting curve. Experiments were performed in triplicate over three biological replicates. (**B, C**) Statistical analyses were performed via unpaired Student's *t*-test. n = 3 biological replicates. ns, not significant; *p < 0.05; **p < 0.01; ***p < 0.001; ****p < 0.0001.

The online version of this article includes the following source data for figure 6:

*Figure 6 continued on next page*

*Figure 6 continued*

**Source data 1.** Unedited gel image of *Figure 6A*.

**Source data 2.** Source data for *Figure 6B, C*.

Ligand binding can stabilize protein structure and lead to a shift in Tm toward higher temperatures. This has been shown previously in the case of UBA5 binding to ATP, where it was determined to interact with a $K_D$ of ~700 µM (*Mashahreh et al., 2018*). The ATP-dependent stabilization of UBA5 allowed us to assess mutational effects on ATP binding using the thermal shift assay. Whereas addition of 5 mM ATP led to a strong, 13°C shift in the Tm of reference UBA5, many of the variants exhibited much weaker stabilization, indicative of a diminished capacity to bind ATP (*Figure 6C*). The other two Group III variants and one Group II variant p.Tyr53Phe showed the strongest defect, with only 4–5°C shifts in Tm upon addition of ATP. The Group II variants p.Met57Val and p.Val260Met as well as one Group IB variant p.Gln312Leu showed a milder effect with a 6–8°C shift in Tm. One Group IB variant p.Arg72Cys and all Group IA variants showed similar ATP-dependent stabilization to reference UBA5. Unlike the melting trend observed in the absence of ATP, in the presence of ATP the p.Gln312Leu variant displayed a single unfolding profile, suggesting that ATP binding corrected the local instability caused by the substitution.

These results show that not all tested variants strongly affect the stability of UBA5 protein. However, many variants impair the ATP-binding capability of the protein. The levels of impairment correlate well with the phenotypic observations in vivo (*Tables 3 and 4*), suggesting that the defect in ATP binding is a major contributor to the LoF associated with the variants. Consistent with their severe LoF in vivo, variants p.Gly168Glu and p.Cys303Arg exhibited impaired protein folding and could not be included in our in vitro analyses.

## Visualizing mutational effects on UFM1 activation and transthiolation

Previous functional characterization of UBA5 activity has relied upon gel-based assays that often lack kinetic information and are less sensitive to subtle changes. We sought to address this problem by developing a real-time, fluorescence polarization (FP) assay for UBA5 activity based upon a method we coined 'UbiReal' (*Franklin and Pruneda, 2019*; *Franklin and Pruneda, 2023*). The approach leverages the large changes in molecular weight that occur as UFM1 is activated by UBA5 and transferred to UFC1 (*Figure 7A*), which are read out as changes in FP of fluorescently labeled UFM1. Indeed, upon addition of UBA5 we observed a large shift in FP of Alexa488-labeled UFM1, which reached

**Table 4.** Summary of protein stability and functions of UBA5 variants.

|  | Variants | Thermal stability | ATP-binding defect [‡] | UFM1 activation | UFM1 transthiolation |
|---|---|---|---|---|---|
| Group IA | p.Ala371Thr | Decreased | No defect | Normal | Decreased |
|  | p.Asp389Gly | Normal | No defect | Decreased | Decreased |
|  | p.Asp389Tyr | Normal | No defect | Decreased | Normal |
| Group IB | p.Arg72Cys | Decreased | No defect | Decreased | Normal |
|  | p.Gln312Leu | Local destabilization[†] | No defect | Normal | Normal |
| Group II | p.Tyr53Phe | Normal | Strong | Decreased | N.A. |
|  | p.Met57Val | Normal | Intermediate | Normal | Normal |
|  | p.Val260Met | Decreased | Intermediate | Decreased | N.A. |
| Group III | p.Arg55His | Normal | Strong | Near baseline | N.A. |
|  | p.Gly168Glu | N.A.* | N.A. | N.A. | N.A. |
|  | p.Leu254Pro | Decreased | Strong | Near baseline | N.A. |
|  | p.Cys303Arg | N.A.* | N.A. | N.A. | N.A. |

*Insoluble in protein purification.
[†]See Results and *Figure 6B*.
[‡]Strong, 4–5°C in Tm shift; intermediate, 6–8°C in Tm shift. Reference UBA5 shows 13 °C in Tm shift upon ATP addition.

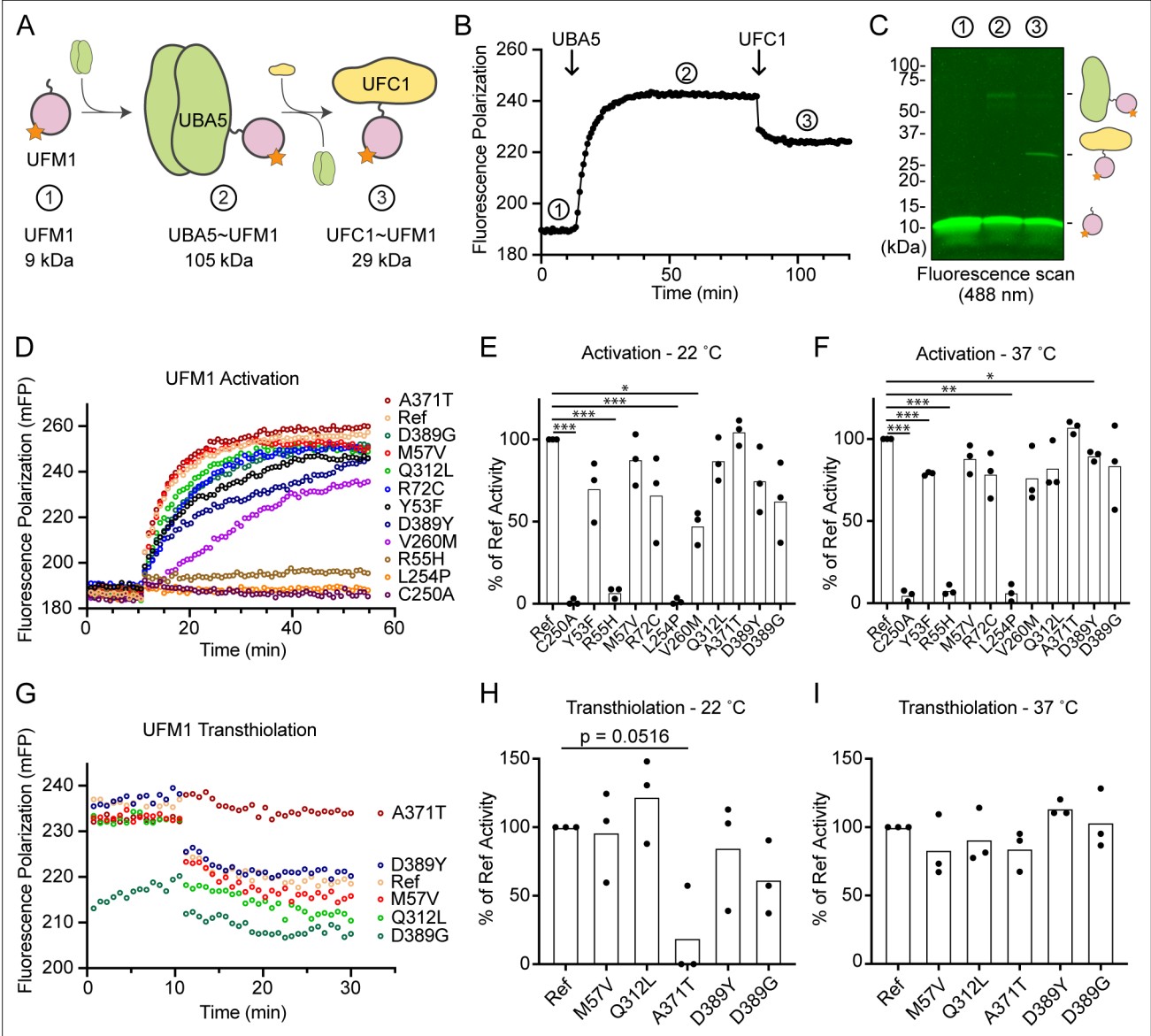

**Figure 7.** Measuring ubiquitin fold modifier 1 (UFM1) activation and transthiolation with UbiReal. (**A**) Cartoon schematic illustrating the complexes formed during UFM1 activation and transthiolation, as well as their expected molecular weights. The fluorescent group attached to UFM1 is denoted by an orange star. The expected molecular weight for the UBA5–UFM1 intermediate is based on a UBA5 homodimer with one UFM1 molecule. (**B**) Proof-of-concept UbiReal assay monitoring the fluorescence polarization (FP) of Alexa488-labeled UFM1 alone (species 1), following addition of UBA5 (species 2), and following addition of UFC1 (species 3). (**C**) Fluorescence scan of samples described in (**B**) separated by sodium dodecyl sulfate–polyacrylamide gel electrophoresis (SDS–PAGE), illustrating the formation of activated UFM1 complexes. Each species is labeled with the analogous cartoon schematic presented in (**A**). (**D**) UbiReal assay tracking UFM1 activation by reference and variant UBA5 proteins over time. (**E**) Area under the curve quantification of UFM1 activation performed at 22°C. Experiments were performed in triplicate over three biological replicates. Statistical analyses were performed using a Welch's *t*-test with comparison to the reference UBA5 data. (**F**) As in (**E**), for reactions performed at 37°C. (**G**) UbiReal assay tracking UFM1 transthiolation for reference UBA5 and variants that showed little or no effect on activation. (**H**) Area under the curve quantification of UFM1 transthiolation performed at 22°C. Experiments were performed in triplicate over three biological replicates. Statistical analyses were performed using a Welch's *t*-test with comparison to the reference UBA5 data. (**I**) As in (**H**), for reactions performed at 37°C. (**E, F, H, I**) Statistical analyses were performed via Welch's *t*-test. n = 3 biological replicates. *$p < 0.05$; **$p < 0.01$; ***$p < 0.001$.

The online version of this article includes the following source data for figure 7:

**Source data 1.** Unedited gel image of *Figure 7C*.

**Source data 2.** Source data for *Figure 7B, E, F, H, I*.

a plateau over the course of ~20 min under these conditions (*Figure 7B*). Addition of UFC1 to the reaction caused a concomitant downward shift in FP that reached a new plateau within ~5 min. To validate the molecular species observed in this assay, we ran samples from each stage of the reaction on non-reducing sodium dodecyl sulfate–polyacrylamide gel electrophoresis (SDS–PAGE) and were able to visualize free UFM1, UBA5–UFM1, and UFC1–UFM1 as predicted (*Figure 7C*).

With the UFM1 UbiReal assay in hand, we proceeded to assess the effects of UBA5 variant on the first step of the reaction: UFM1 activation. As anticipated, addition of reference UBA5 caused a rapid shift in FP over time, whereas the enzyme-dead p.Cys250Ala variant remained at baseline (*Figure 7D*). We quantified these kinetic data with area under the curve (AUC) analysis using data collected just before addition of UBA5 to establish a baseline. The panel of UBA5 variants displayed a wide range of effects. The two Group III variants p.Arg55His and p.Leu254Pro remained near the baseline, indicating a severe impairment in UFM1-activating function. An intermediate, statistically significant effect was observed with the p.Val260Met variant (Group II), while the remaining variants showed mild or, in the case of p.Ala371Thr (Group IA), no decrease in the rate of UFM1 activation (*Figure 7E*). Similar trends were observed at both 22 and 37°C, though interestingly the effect of p.Val260Met substitution was less severe at higher temperature (*Figure 7F*). The results show that a defect in UFM1 activation contributes to the LoF in all variants except for p.Ala371Thr. The severity of LoF in UFM1 activation correlates with our phenotypic observations in vivo (*Tables 3 and 4*).

Having observed defects in UFM1 activation for many of the UBA5 variants, we next analyzed those that exhibited mild or no effect for their ability to complete the second enzymatic role of UBA5: UFM1 transthiolation onto UFC1. After forming the activated UBA5–UFM1 intermediates, we added UFC1 to the reactions and monitored the decay in FP over time (*Figure 7G*). Though many of the UBA5 variants showed similar trends for transthiolation, the p.Ala371Thr variant immediately stood out for having no effect on UFM1 activation (*Figure 7D–F*), but a greatly impaired ability to transfer UFM1 onto UFC1 (*Figure 7G*). We quantified the kinetic data using AUC analysis of the inverted curve and observed a remarkable defect in the ability of the p.Ala371Thr variant to catalyze UFM1 transthiolation (*Figure 7H*). Interestingly, this defect is much more pronounced at 22°C than at 37°C (*Figure 7I*). These results suggest that aside from a minor decrease in UBA5 stability (*Figure 6B*), UFM1 transthiolation is the only possible defect caused by the p.Ala371Thr variant. Transthiolation could be defective in other variants as well, such as p.Leu254Pro (Group III), but this effect is likely overshadowed by upstream effects on UBA5 stability, ATP binding, and/or UFM1 activation.

## Discussion

In this study, we assessed the strength and properties of variants identified in individuals with DEE44 using a humanization strategy in fruit flies. Germline knockout of *Uba5* causes embryonic lethality in both flies (this study) and mice (*Tatsumi et al., 2011*), and no individuals have been identified with biallelic null variants in humans. The lethality of *Uba5* knockout mice is caused by hematopoietic defects (*Tatsumi et al., 2011*), however, the clinical presentations in DEE44 patients are predominantly related to the CNS (*Arnadottir et al., 2017*; *Briere et al., 2021*; *Colin et al., 2016*; *Daida et al., 2018*; *Low et al., 2019*; *Mignon-Ravix et al., 2018*; *Muona et al., 2016*). These findings suggest that DEE44 is caused by partial LoF of *UBA5*, and that *Uba5* knockout models are not suitable to study disease pathogenesis. We generated a *Uba5*^T2A-Gal4 allele that corresponds to a severe LoF allele but also leads to expression of GAL4 in the same spatial and temporal expression pattern of *Uba5*, which can then drive expression of the human *UBA5* cDNA (*Figure 2B*). Expression of reference *UBA5* cDNA fully rescues the loss of *Uba5*, showing that the proteins are functionally similar. This allowed us to assess the properties of the known *UBA5* variants and establish an allelic series in vivo. We measured an array of phenotypes in flies that relate to the phenotypes observed in affected individuals, including developmental delay, motor defects, and bang sensitivity (seizure-like behavior). Based on the in vivo data, we establish groups of variants with different strengths from least severe to most severe: Groups IA, IB, II, and III (*Table 3*). To facilitate the discussion, hereafter we also refer to the deletion, nonsense, frameshift, and splicing variants observed in affected individuals as Group IV variants, although they were not functionally tested in this study.

To correlate the allelic strength with the functional defects caused by the variants, we first examined the stability of UBA5 variants and assessed their activity in three key steps involved in UBA5 enzyme function: ATP binding, UFM1 activation, and UFM1 transthiolation. Two variants were not tested due

to instability during protein purification, while all remaining tested variants exhibited defects in at least one assay, suggesting that they cause protein LoF (*Table 4*).

Group IA includes three variants, p.Asp389Gly, p.Asp389Tyr, and p.Ala371Thr. The three variants fully rescued the defects caused by the loss of *Uba5* in flies, indicating that they do not affect UBA5 function or only cause a mild LoF of UBA5. The biochemical assays show that p.Asp389Gly and p.Asp389Tyr only cause mild, but statistically insignificant defects in UFM1 activation, consistent with the complete rescue of the loss of *Uba5* in humanized flies (*Figure 4*). The other variant, p.Ala371Thr, has a minor allele frequency (MAF) of 0.0019 (517/274,744 alleles) in general population and is more frequent in Finnish population (MAF = 0.0059, 149/24,996 alleles) in gnomAD v2.1.1. It is also the most commonly observed variant in affected individuals (*Table 1*). Individuals homozygous for this variant have been identified in the Finnish and Icelandic population but they do not present with obvious symptoms related to DEE44 (*Arnadottir et al., 2017*; *Colin et al., 2016*; *Muona et al., 2016*), consistent with our observations in flies. Interestingly, although the biochemical assays showed no dramatic effect of Ala371Thr on UBA5 stability, ATP binding, or UFM1 activation, the transthiolation of UFM1 onto UFC1 is impaired. While it may seem intuitive that defective UFM1 activation or transthiolation would be equally detrimental, formation of the activated UBA5–UFM1 intermediate is the rate-limiting step of the reaction and hence more sensitive to perturbations (*Gavin et al., 2014*). Importantly, the defect is observed at 22°C but not at 37°C in vitro, indicating that p.Ala371Thr is a very weak allele, consistent with all the in vivo and human genetics data.

Groups IB and II include variants that have an intermediate effect on the protein function in vivo. They rescued the lethality of *Uba5* mutant flies, but cause phenotypes of various levels of severity in adult flies. In biochemical assays, Group II variants exhibit a decreased capacity to bind ATP and a mild loss in the ability to activate UFM1, though the limitations of our assay precluded statistically significant effects among several variants. In Group IB, the p.Arg72Cys only causes a mild defect in UFM1 activation, consistent with it being near but not directly involved in ATP binding. The novel Group 1B variant reported herein, p.Gln312Leu, exhibited virtually no biochemical phenotype aside from signs of local instability in the UBA5 structure, which is stabilized in the presence of ATP. Among the sites of Group IB and II variants, only the Val260 residue lies at the homodimeric interface while no others lie immediately at a key interface for UBA5 function. This is consistent with p.Val260Met being the only variant among this set that exhibited severe activity defects in vitro. Despite this, all Group IB and II variants caused phenotypes in flies. It is possible that the subtle effects of these substitutions are exacerbated in certain cellular conditions, such as ER stress.

Group III variants failed to rescue the lethality of the *Uba5* mutants, suggesting that they correspond to severe LoF alleles. This group includes four variants, p.Arg55His, p.Gly168Glu, p.Leu254Pro, and p.Cys303Arg. Two of the affected sites, Gly168 and Cys303, are buried in the UBA5 structure. Interestingly, neither of these variants are soluble when produced in *E. coli*, indicating compromised protein folding. The two remaining variants, p.Arg55His and p.Leu254Pro are in very close proximity of the UBA5 active site (*Figure 5B*) and may affect ATP binding and cause decreased conformational dynamics of the active site Cys. Both variants show a diminished capacity to bind ATP, and are incapable of activating UFM1, consistent with the observations that they are very severe LoF alleles in the fly. Hence, in Group III two alleles severely disrupt UBA5 protein stability while two others affect enzyme catalysis.

Using our variant classification, we retrospectively analyzed the allelic combinations in reported DEE44 cases. The most severely affected individual is homozygous for a Group II variant p.Tyr53Phe. However, this individual was from a consanguineous family so other variants may correspond to the phenotypic presentation of this individual (*Cabrera-Serrano et al., 2020*). Most (21/25) individuals are compound heterozygous for one allele from Group IA or IB and another allele from Group III or IV (*Table 1*). This strongly suggests that the pairing of a mild LoF with a severe LoF allele is required to allow individuals to survive and manifest the disease. However, two individuals from a previous report (IA/II) and this study (IB/II) show that the disease is also associated with a combination of two partial LoF alleles (*Table 1*). Finally, no affected individual is free of alleles from Group II/III/IV, indicating that combinations of IA/IB alleles may not cause disease. This is also supported by the observation that homozygous p.Ala371Thr (Group IA) individuals are not affected (see above) (*Arnadottir et al., 2017*; *Colin et al., 2016*; *Muona et al., 2016*). Our results provide compelling evidence for the interpretation of existing and future variants as well as the prediction of pathogenicity of allelic combinations in

clinical genetic analyses, especially considering the very limited number of reported individuals and the partial LoF mechanism of the disease.

Three other genes in the UFMylation pathway are associated with diseases that share symptoms with DEE44. Variants in *UFSP2* (OMIM: #611482) cause another DEE (DEE106, OMIM: #620028) (*Ni et al., 2021*). *UFM1* (OMIM: #610553) is associated with hypomyelinating leukodystrophy 14 (HLD14, OMIM: #617899) (*Hamilton et al., 2017*; *Nahorski et al., 2018*). Variants in *UFC1* (OMIM: #610554) cause a neurodevelopmental disorder with spasticity and poor growth (NEDSG, OMIM: #618076) (*Nahorski et al., 2018*). All these disorders cause global developmental delay, hypotonia, spasticity, seizures, delayed myelination, and microcephaly, consistent with them sharing a similar etiology. The UbiReal system developed herein can be easily adapted to test the variants in these other UFMylation genes in future work. In addition, given that the genes are also highly conserved in flies, the allelic strengths of the variants in these genes could also be established in vivo using our fruit fly model.

## Materials and methods

### Human genetics

The proband was recruited through the Doernbecher Children's Hospital and informed consent was obtained from legal guardians of the proband. The human study was approved by the Institutional Review Board at Oregon Health & Science University (Title: Pediatric *UBA5* patient case study; Investigator: Ruth Napier; IRB ID: STUDY00025283). The legal guardians of the proband consent to have the results of this research work published.

Trio exome sequencing was conducted by GeneDx using DNA extracted from blood. DNA was enriched using a proprietary capture system developed by GeneDx for next-generation sequencing. The enriched targets were simultaneously sequenced with paired-end reads on an Illumina platform. Bidirectional sequence reads were assembled and aligned to Human Genome Sequencing Center (HGSC) build 37, human reference genome 19. Reported variants were confirmed, if necessary, by an appropriate orthogonal method.

### *Drosophila* strains and genetics

All fruit fly strains used in this study were cultured using standard fly food in a 25°C incubator unless a different culturing temperature was specifically indicated. The $Uba5^{KO}$ mutant and the human *UAS-UBA5* transgenic fly lines were generated in the Bellen lab (for methods, see below). The $Uba5^{T2A-Gal4}$ (#78928), *UAS-mCherry.nls* (#38424), *UAS-FLP* (#4540), $Uba5^{GR}$ (#30359), *da-Gal4* (#5460), *Act-Gal4* (#4414), *elav-Gal4* (#8765), and *repo-Gal4* (#7415) lines were obtained from the Bloomington *Drosophila* Stock Center (BDSC).

### Generation of $Uba5^{KO}$ allele

The $Uba5^{KO}$ allele was generated using CRISPR/Cas9 genome engineering technology as previously described (*Port et al., 2014*). A TKO transgenic line expressing a *Uba5*-targeting single-guide RNA (sgRNA) is available from the BDSC (BDSC #81448) (*Zirin et al., 2020*). The sgRNA (CAATCCGT ACAGCCGCCTGA) targets the coding region in the Exon 1 of the only *Uba5* transcript. To generate indel variants, the TKO flies were crossed to *nos-Cas9* transgenic flies (BDSC #78781) and then the first generation (F1) female progenies carrying both sgRNA and Cas9 were crossed with first chromosome balancer flies. Single F2 females were crossed with balancer flies again to establish ~20 individual stocks with potential indel variants. Due to the lethality of *Uba5* mutants, the stocks without unbalanced flies were screened for *Uba5* indel variant by genomic PCR and Sanger sequencing. One mutant line with NM_132494.3 (*Uba5*):c.164_174del (p.Arg55ProfsTer87) varaint was isolated and designated as $Uba5^{KO}$ allele in this study.

### Generation of UAS-UBA5 transgenic stocks

Human *UAS-UBA5* transgenic fly lines were generated as previously described (*Harnish et al., 2019*). In brief, the human *UBA5* cDNA sequences were cloned into the pGW-UAS-HA.attB vector (*Bischof et al., 2013*) using the Gateway Cloning system (Thermo Fisher) and validated by sequencing. The cDNA vectors were then injected into fly embryos and inserted into the VK37 (BDSC #24872) docking site by φC31-mediated transgenesis (*Venken et al., 2006*). The human *UBA5* cDNA clone

corresponding to Genebank transcript NM_024818.6 was obtained from the Ultimate ORF Clones library (Thermo Fisher). The *UBA5* variants were introduced into the reference cDNA using Q5 site-directed mutagenesis (NEB) before the cDNA was cloned into the pGW-UAS-HA.attB vector.

## Immunostaining and confocal microscopy

Larval and adult flies were dissected in 1× phosphate-buffered saline and the specimens were processed for immunostaining. Briefly, the tissues were fixed in 4% paraformaldehyde followed by normal goat serum blocking and incubation in the primary antibody (Rat anti-Elav, Developmental Studies Hybridoma Bank (DSHB) #7E8A10, 1:500; Mouse anti-Repo, DSHB #8D12, 1:50; Mouse anti-CSP, DSHB #6D6, 1:200; FITC-conjugated anti-HRP, Jackson ImmunoResearch, 1:200). Cy5-conjugated secondary antibodies were used to detect the primary antibodies. Samples were mounted on slides using RapidClear (SUNJin Lab) and the images were captured using a confocal microscope (Zeiss 710). For NMJ morphology analysis, the NMJs from larval muscle 4 of abdominal segments 2, 3, or 4 were imaged and the bouton numbers and sizes were quantified using ImageJ software.

## *Drosophila* lifespan assay

For the measurement of lifespan, freshly eclosed flies were collected in separate vials and maintained at 25°C. Flies were transferred every day to fresh food in the first 6 days and every other day afterward. Survival was determined during every transfer. The results are represented as Kaplan–Meier curves.

## *Drosophila* behavioral assay

To measure negative geotaxis, flies were transferred to a clean vial for at least 20 min prior to the experiment. During the test, flies were tapped to the bottom of the vial and their negative geotaxis climbing ability was measured. In each measurement, flies were allowed to climb for 30 s, after which the climbing distances were measured (18 cm is maximum). To perform bang-sensitive paralytic analyses, adult flies were transferred to a clean vial and vortexed at maximum speed for 10 s, after which the time required for flies to stand on their feet was counted (30 s is maximum).

## Protein expression and purification

The reference UBA5 gene and UFM1 cloned into pET15b and UFC1 cloned into pET32a were kind gifts from R. Wiener (The Institute for Medical Research Israel-Canada). The UBA5 p.Met57Val and p.Gln312Leu substitutions were cloned into this background by Quikchange PCR using Phusion DNA polymerase. All other UBA5 variant substitutions were subcloned into pOPIN-B using the constructs described above as templates. All of these constructs encoded N-terminal His-tags, and were purified in a similar manner. After plasmid transformation into *E. coli* Rosetta (DE3) cells (MiliporeSigma), cultures were grown at 37°C in Luria Broth containing 35 µg/ml chloramphenicol and 50 µg/ml of kanamycin. Once an optical density (600 nm) between 0.4 and 0.6 was reached, cultures were cooled to 18°C and protein expression was induced with 0.5 mM isopropyl β- d-1-thiogalactopyranoside (IPTG). Cells were harvested by centrifugation after 24 hr of expression and resuspended in 50 mM $NaPO_4$, 500 mM NaCl, 2 mM β-mercaptoethanol, pH 8.0 (Buffer A). The cell pellet was then subjected to a freeze–thaw cycle before adding DNase, phenylmethylsulfonyl fluoride (PMSF), lysozyme, and SigmaFAST Protease Inhibitor Cocktail (MiliporeSigma) and allowed to incubate on ice for 30 min. The cell pellets were then lysed by either french press or sonication, depending on the volume to be lysed. Lysates were then centrifuged at 35,000 × *g*, and the clarified lysate was added to a column containing HisPur Cobalt affinity resin (Thermo Fisher), allowed to bind for 10 min, and washed with 1 l of Buffer A + 10 mM Imidazole. Proteins were then eluted with 5 ml of Buffer A + 350 mM imidazole in a stepwise manner for a total elution volume of 25 ml. Purity of the fractions was analyzed via SDS–PAGE and those with highest purity were pooled and dialyzed overnight into 25 mM Tris, 100 mM NaCl, 2 mM dithiothreitol (DTT), pH 8.0 (Buffer B) at 4°C. After 24 hr of dialysis, UBA5 proteins were further purified on a RESOURCE Q (Cytivia) anion exchange chromatography column equilibrated in Buffer B. The protein was eluted over a 20 column volume gradient against Buffer B + 1 M NaCl. Peak protein fractions were pooled and concentrated using Amicon centrifugal filters (MiliporeSigma) before being applied to a HiLoad Superdex 75 16/600 pg size exclusion chromatography column (Cytivia) equilibrated in Buffer B. Following affinity purification, the His-tags of UFM1 and UFC1 were removed by TEV cleavage during overnight dialysis into Buffer B at 4°C. Proteins were

then concentrated using Amicon centrifugal filters and applied to a HiLoad Superdex 75 16/600 pg as above. Peak fractions were evaluated for purity via SDS–PAGE, pooled, and concentrated before being quantified by absorbance at 280 nm and flash frozen above 10× their working stock concentration. All protein samples were stored at −80°C.

### Fluorescence-based UBA5 activity assays

UFM1-Alexa 488 substrates were prepared using an Alexa Fluor 488 TFP ester (Thermo Fisher). Labeling was performed at room temperature for 1 hr in 0.1 M sodium bicarbonate buffer at pH 7.5, which directs labeling toward the N-terminus. Following labeling, excess fluorophore was quenched with addition of 150 mM Tris pH 7.4 and separated by size exclusion chromatography on a HiLoad Superdex 75 16/600 pg as above. Transfer and activation of UFM1 onto UBA5 and further transthiolation to UFC1 were monitored by FP based on published methods from the ubiquitin system (*Franklin and Pruneda, 2019*). FP was measured using a BMG LabTech CLARIOstar plate reader at an excitation wavelength of 482 nm, an LP 504 nm dichroic mirror, and an emission wavelength of 530 nm. Free UFM1 was used as a reference with a target FP of 190. All assays were performed in black Greiner 384-well small-volume HiBase, low protein-binding microplates. UFM1 and UBA5 stocks were prepared at 2× assay conditions in 25 mM $NaPO_4$, 150 mM NaCl, 10 mM $MgCl_2$, pH 7.4 (Buffer C). UFM1 was prepared at 100 nM in Buffer C + 20 mM ATP (pH 7); UBA5 variants were prepared at 1 µM in Buffer C. UFC1 was prepared at 3 µM (10× assay concentration) in Buffer C. Final assay conditions in 20 µl volumes were 50 nM UFM1, 500 nM UBA5 variants, and 300 nM UFC1 in Buffer C + 10 mM ATP. FP data were first collected for the substrate only to establish a baseline before addition of UBA5 variants at 1:1 ratio to reach the described assay conditions. FP values were collected in 45-s intervals until the reference UBA5 readings plateaued, at which point UFC1 was added at 1:10 ratio to achieve final assay conditions. Reactions were performed in triplicate for each of three experimental replicates, both at 22 and 37°C. AUC calculations were performed using Prism 9.5, with baseline values calculated from data collected prior to addition of UBA5/UFC1. For the UFM1 activation stage, AUC was determined over a 45-min window following addition of UBA5. For the UFM1 transthiolation stage, AUC was determined over a 20-min window following addition of UFC1.

### Gel-based UBA5 activation and transthiolation assay

Conditions for the gel-based assay were identical to those described above for the 22°C FP assays. Parallel reactions were prepared, one to be read out by FP using the CLARIOstar plate reader and the other was left in a low light environment at room temperature. After 15 min of establishing a baseline FP reading for UFM1 alone, a 20-µl sample was taken from the parallel reaction and quenched using non-reducing sample buffer. FP readings were paused and reference UBA5 was added to both reactions before continuing FP data collection. The FP values were allowed to plateau before another 20 µl sample was collected from the parallel reaction and quenched with non-reducing sample buffer. FP readings were then paused and UFC1 was added to both reactions before continuing FP data collection. The FP values were allowed to plateau before taking a final 20 µl sample from the parallel reaction and quenching with non-reducing sample buffer. Gel samples were run on a TGX 4–20% SDS–PAGE gradient gel (Bio-Rad). The resulting gel was imaged using a Sapphire Biomolecular Imager (Azure Biosystems). This experiment was performed in triplicate.

### Thermal shift assay

The thermal shift assay was conducted in MicroAmp Fast 96-Well Reaction Plates (Applied Biosystems) with SYPRO Orange Protein Gel Stain (MiliporeSigma) using a QuantStudio 3 Real-Time PCR system (Applied Biosystems). Assays were performed in 20 µl volumes containing 5 µM UBA5 variants and 20× SYPRO dye (diluted from a 5000× stock) in 25 mM $NaPO_4$, 150 mM NaCl, 10 mM $MgCl_2$, pH 7.4 with or without 5 mM ATP. The protocol ramped temperature from 22 to 99°C over a gradient of 0.1°C every 5 s, and fluorescence was monitored using an excitation wavelength of 580 ± 10 nm and an emission wavelength of 623 ± 14 nm.

### Statistical analysis

Statistical analyses were carried out using the Student's unpaired two-tailed *t*-test for comparison of two groups or the Welch's *t*-test for data normalized to reference UBA5. Multiple comparisons within

the group were tested against the corresponding control. Kaplan–Meier survival curves were analyzed using Gehan–Breslow–Wilcoxon test and log-rank test. Calculated p values of less than 0.05 were considered significant. All statistical analyses were performed using GraphPad Prism, version 9.5.0 (GraphPad Software).

## Materials availability

All materials newly created in this paper are available from the correspondence authors upon request.

## Web resources

OMIM, https://omim.org/
DIOPT, https://www.flyrnai.org/cgi-bin/DRSC_orthologs.pl/
gnomAD, https://gnomad.broadinstitute.org/
CADD, https://cadd.gs.washington.edu/
SIFT, https://sift.bii.a-star.edu.sg/
PolyPhen2, http://genetics.bwh.harvard.edu/pph2/
MutationTaster, https://www.mutationtaster.org/
PROVEAN, http://provean.jcvi.org/

## Acknowledgements

We thank the proband and his family for agreeing to participate in this study. We thank the Bellen and Yamamoto lab members for their discussion and suggestions in this study. We thank Ms. Hongling Pan for the injection of transgenic fly lines. We thank the BDSC for fly stocks, the DSHB for antibodies, and R Wiener (The Institute for Medical Research Israel-Canada) for sharing plasmids. HJB, OK, and SY were supported by the Office of Research Infrastructure Programs (ORIP) of the NIH (award U54 OD030165). HJB was also supported by the ORIP of the NIH (awards R24 OD022005 and R24 OD031447), the Huffington Foundation, and the Jan & Dan Duncan Neurological Research Institute at Texas Children's Hospital. The work was also supported by the Baylor College of Medicine IDDRC P50HD103555 from the Eunice Kennedy Shriver National Institute of Child Health and Human Development for use of the Microscopy Core facilities. JNP and RJN were supported by the OHSU Molecular Microbiology and Immunology Interdisciplinary Pilot Award and the Oregon Clinical and Translational Research Institute's Biomedical Innovation Program NCATS UL1TR002369 from the NIH. JNP was also supported by an NIGMS R35 grant (R35 GM142486), and RJN was also supported by a VA CDA2 grant (5IK2BX004523). DAS and LCB were supported by the NIH common fund through the Office of Strategic Coordination/Office of the NIH Direction (award U01 HG007690), the Hill Family Fund for the Diagnosis, Management of Rare and Undiagnosed Diseases at Mass General, and American Institute for Neuro Integrative Development Inc (AIND).

## Additional information

### Competing interests

Hugo J Bellen: Reviewing editor, eLife. The other authors declare that no competing interests exist.

### Funding

| Funder | Grant reference number | Author |
| --- | --- | --- |
| Office of Research Infrastructure Programs, National Institutes of Health | U54 OD030165 | Oguz Kanca<br>Shinya Yamamoto<br>Hugo J Bellen |
| Office of Research Infrastructure Programs, National Institutes of Health | R24 OD022005 | Hugo J Bellen |

| Funder | Grant reference number | Author |
|---|---|---|
| Office of Research Infrastructure Programs, National Institutes of Health | R24 OD031447 | Hugo J Bellen |
| Huffington Foundation | | Hugo J Bellen |
| Eunice Kennedy Shriver National Institute of Child Health and Human Development | P50 HD103555 | Hugo J Bellen |
| OHSU Molecular Microbiology and Immunology Interdisciplinary Pilot Award | | Ruth J Napier Jonathan N Pruneda |
| National Center for Advancing Translational Sciences | UL1 TR002369 | Ruth J Napier Jonathan N Pruneda |
| National Institute of General Medical Sciences | R35 GM142486 | Jonathan N Pruneda |
| U.S. Department of Veterans Affairs | CDA2 grant 5IK2BX004523 | Ruth J Napier |
| Office of Strategic Coordination | U01 HG007690 | Lauren C Briere David A Sweetser |
| Hill Family Fund for the Diagnosis | | Lauren C Briere David A Sweetser |
| Management of Rare and Undiagnosed Diseases at Mass General | | Lauren C Briere David A Sweetser |
| American Institute for Neuro Integrative Development Inc | | Lauren C Briere David A Sweetser |

The funders had no role in study design, data collection, and interpretation, or the decision to submit the work for publication.

## Author contributions

Xueyang Pan, Conceptualization, Data curation, Investigation, Visualization, Methodology, Writing – original draft, Project administration, Writing – review and editing; Albert N Alvarez, Michael W Crawford, Data curation, Investigation, Methodology; Mengqi Ma, Shenzhao Lu, Data curation, Investigation; Lauren C Briere, Resources, Funding acquisition, Investigation, Writing – review and editing; Oguz Kanca, Resources, Funding acquisition, Investigation, Methodology, Writing – review and editing; Shinya Yamamoto, Resources, Supervision, Funding acquisition, Project administration, Writing – review and editing; David A Sweetser, Resources, Supervision, Funding acquisition, Writing – review and editing; Jenny L Wilson, Resources, Investigation, Writing – review and editing; Ruth J Napier, Funding acquisition, Project administration, Writing – review and editing; Jonathan N Pruneda, Hugo J Bellen, Conceptualization, Resources, Supervision, Funding acquisition, Methodology, Writing – original draft, Project administration, Writing – review and editing

## Author ORCIDs

Xueyang Pan ⓘ https://orcid.org/0000-0003-4453-4971
Shenzhao Lu ⓘ http://orcid.org/0000-0003-3117-3900
Michael W Crawford ⓘ https://orcid.org/0000-0001-7779-4636
David A Sweetser ⓘ http://orcid.org/0000-0002-1621-3284
Jonathan N Pruneda ⓘ https://orcid.org/0000-0002-0304-4418
Hugo J Bellen ⓘ https://orcid.org/0000-0001-5992-5989

### Ethics

The proband was recruited through the Doernbecher Children's Hospital and informed consent was obtained from legal guardians of the proband. The human study was approved by the Institutional Review Board at Oregon Health & Science University (Title: Pediatric UBA5 patient case study; Investigator: Ruth Napier; IRB ID: STUDY00025283). The legal guardians of the proband consent to have the results of this research work published.

Reviewer #1 (Public Review): https://doi.org/10.7554/eLife.89891.3.sa1
Reviewer #2 (Public Review): https://doi.org/10.7554/eLife.89891.3.sa2
Reviewer #3 (Public Review): https://doi.org/10.7554/eLife.89891.3.sa3
Author Response https://doi.org/10.7554/eLife.89891.3.sa4

---

## Additional files

### Supplementary files

• MDAR checklist

• Supplementary file 1. Summarizes the clinical features of all reported individuals with *UBA5*-associated developmental and epileptic encephalopathy 44 (DEE44) as well as the proband reported in this study.

### Data availability

All data generated or analyzed during this study are included in the manuscript and supporting file. Source data files have been provided for Figures 4, 6, and 7.

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
