## [Editor Report · eLife assessment]

The authors establish a *Drosophila* model to assess the severity of disease-linked alleles of Uba5. Using both in vivo and in vitro experiments, this **valuable** study demonstrates the alleles fall into mild, intermediate, and severe classes, with **convincing** evidence to support their conclusion. This well-executed study establishes a model for further characterization of Uba5-related phenotypes in a powerful model system.

---

## [Referee Report · Reviewer #1 (Public Review)]

Summary:

In this study, the authors generate a *Drosophila* model to assess disease-linked allelic variants in the UBA5 gene. In humans, variants in UBA5 have been associated with DEE44, characterized by developmental delay, seizures, and encephalopathy. Here, the authors set out to characterize the relationship between 12 disease-linked variants in UBA5 using a variety of assays in their *Drosophila* Uba5 model. They first show that human UBA5 can substitute all essential functions of the *Drosophila* Uba5 ortholog, and then assess phenotypes in flies expressing the various disease variants. Using these assays, the authors classify the alleles into mild, intermediate, and severe loss-of-function alleles. Further, the authors establish several important in vitro assays to determine the impacts of the disease alleles on Uba5 stability and function. Together, they find a relatively close correlation between in vivo and in vitro relationships between Uba5 alleles and establish a new *Drosophila* model to probe the etiology of Uba5-related disorders.

Strengths:

Overall, this is a convincing and well-executed study. There is clearly a need to assess disease-associated allelic variants to better understand human disorders, particularly for rare diseases, and this humanized fly model of Uba5 is a powerful system to rapidly evaluate variants and relationships to various phenotypes. The manuscript is well written, and the experiments are appropriately controlled.

---

## [Referee Report · Reviewer #2 (Public Review)]

Relative simplicity and genetic accessibility of the fly brain makes it a premier model system for studying the function of genes linked to various diseases in humans. Here, Pan et al. show that human UBA5, whose mutations cause developmental and epileptic encephalopathy, can functionally replace the fly homolog Uba5. The authors then systematically express in flies the different versions of the gene carrying clinically relevant SNPs and perform extensive phenotypic characterization such as survival rate, developmental timing, lifespan, locomotor and seizure activity, as well as in vitro biochemical characterization (stability, ATP binding, UFM-1 activation) of the corresponding recombinant proteins. The biochemical effects are well predicted by (or at least consistent with) the location of affected amino acids in the previously described Uba5 protein structure. Most strikingly, the severity of biochemical defects appears to closely track the severity of phenotypic defects observed in vivo in flies. While the paper does not provide many novel insights into the function of Uba5, it convincingly establishes the fly nervous system as a powerful model for future mechanistic studies.

One potential limitation is the design of the expression system in this work. Even though the authors state (ln. 127-128) that "human cDNA is expressed under the control of the endogenous Uba5 enhancer and promoter", it is in fact the Gal4 gene that is expressed from the endogenous locus (which authors also note in the same paragraph 138-139), meaning that the cDNA expression level would inevitably be amplified in comparison. While I do not think this weakens the conclusions of this paper, it may impact the interpretation of future experiments that use these tools. Especially considering the authors argue that most disease variants of UBA5 are partial loss-of-functions, the amplification effect could potentially mask the phenotypes of milder hypomorphic alleles. Temperature sensitivity of Gal4 expression may allow calibrating levels to reduce the impact of this amplification, but the revised manuscript still does not openly acknowledge or discuss this potential caveat.

---

## [Referee Report · Reviewer #3 (Public Review)]

Summary:

Variants in the UBA5 gene are associated with rare developmental and epileptic encephalopathy, DEE44. This research developed a system to assess in vivo and in vitro genotype-phenotype relationships between UBA5 allele series by humanized UBA5 fly models and biochemical activity assays. This study provides a basis for evaluating current and future individuals afflicted with this rare disease.

Strengths:

The authors developed a method to measure the enzymatic reaction activity of UBA5 mutants over time by applying the UbiReal method, which can monitor each reaction step of ubiquitination in real time using fluorescence polarization. They also classified fruit fly carrying humanized UBA5 variants into groups based on phenotype. They found a correlation between biochemical UBA5 activity and phenotype severity.

Weaknesses:

In the case of human DEE44, compound heterozygotes with both loss-of-function and hypomorphic forms (e.g., p.Ala371Thr, p.Asp389Gly, p.Asp389Tyr) may cause disease states. The presented models have failed to evaluate such cases.

---

## [Author Response]

The following is the authors’ response to the original reviews.

We greatly appreciate the valuable and constructive review of our manuscript. The reviewers’ comments have helped us to improve the quality of the paper. Here we provide detailed responses to the reviewers’ comments and discuss the new experiments we performed.

**Reviewer #1**
Summary:In this study, the authors generate a *Drosophila* model to assess disease-linked allelic variants in the UBA5 gene. In humans, variants in UBA5 have been associated with DEE44, characterized by developmental delay, seizures, and encephalopathy. Here, the authors set out to characterize the relationship between 12 disease-linked variants in UBA5 using a variety of assays in their *Drosophila* Uba5 model. They first show that human UBA5 can substitute all essential functions of the *Drosophila* Uba5 ortholog, and then assess phenotypes in flies expressing the various disease variants. Using these assays, the authors classify the alleles into mild, intermediate, and severe loss-of-function alleles. Further, the authors establish several important in vitro assays to determine the impacts of the disease alleles on Uba5 stability and function. Together, they find a relatively close correlation between in vivo and in vitro relationships between Uba5 alleles and establish a new *Drosophila* model to probe the etiology of Uba5-related disorders.Strengths:Overall, this is a convincing and well-executed study. There is clearly a need to assess disease-associated allelic variants to better understand human disorders, particularly for rare diseases, and this humanized fly model of Uba5 is a powerful system to rapidly evaluate variants and relationships to various phenotypes. The manuscript is well written, and the experiments are appropriately controlled.Recommendations For The Authors:1. It would seem of value to determine what tissue(s) the essential function of Uba5 resides. The authors nicely detail the expression of Uba5 in a subset of neurons and glia, and indicate it is expressed in a variety of other tissues. Null mutants are embryonic lethal, suggesting an essential function. From the mouse study cited, it appears Uba5 functions early in development in the hematopoietic system. The authors can express their UAS-Uba5 rescue construct using a variety of tissue-specific Gal4 lines to determine whether the essential function of Uba5 is in the nervous system or other tissues, which would be of interest in understanding key functions of Uba5.

We thank the reviewer for the suggestion. We tried to rescue the lethality of the Uba5 mutants by expressing human UBA5 reference protein in different tissues. We found that ubiquitous expression of UBA5 (da-GAL4 or act-GAL4) successfully rescues the lethality, however, expression of UBA5 in neurons (elav-GAL4), glia (repo-GAL4), or both neurons and glia does not. In addition, expression of UBA5 in fat body (SPARC-GAL4) or muscles (Mef2-GAL4) does not rescue the lethality either. These results suggest that Uba5 is required in multiple tissues in flies. These data are included in the revised manuscript.

2). Do intermediate Uba5 alleles impact synaptic function or growth? The etiology of the disease is linked with epilepsy and neurodevelopmental disorders, and the interesting parallels the authors note between Uba5 and Para expression indicate perhaps shared roles in neurons that drive firing activity. Together, these lines of evidence may suggest the Uba5 alleles may have possible impacts on synaptic growth, morphology, and/or function. It would be of interest to examine the larval neuromuscular junction and assess NMJ growth, morphology, and perform some basic electrophysiology to determine if there are any functional defects.

Following the reviewer’s suggestion, we tested the morphology of NMJs in the humanized flies. We did not observe any obvious changes in the number or size of the synaptic boutons caused by the Group II variants. Hence, we conclude that the Uba5 variants do not cause an obvious defect in synaptic growth. The results are included in the Figure S3.

More generally, can the authors comment on the expression pattern of Uba5? One might consider something like Uba5 to be a "housekeeping" gene and expressed/required in most if not all cell types. From the data presented in Fig. 2, it appears expression is more sparse, perhaps, as the authors point out, because of roles in mature neurons that actively fire (like Para). Are neuronal targets of Uba5 known, which might suggest key pathways it modulates?

We showed that Uba5 is broadly expressed in third instar larvae. FlyAtlas2 and FlyCellAtlas datasets show that Uba5 is broadly expressed but not in all the cells. In the larval CNS and adult brain, Uba5 is not expressed in all cells either. Hence, we cannot say Uba5 is a “housekeeping” gene. Regarding the neuronal targets of Uba5, we do not know which types of neurons express Uba5 and which pathways Uba5 modulates. This could be studied in the future.

1. Does strong overexpression of the various Uba5 alleles in otherwise wild-type flies cause any phenotypes? This might support possible antimorphic/dominant negative functions of some of the variants. Is it plausible that any of the alleles could impact oligomerization of Uba5?

We have not observed compromised viability or any obvious phenotype in flies overexpressing human reference UBA5 or UBA5 variants. So, our results do not support a dominant negative effect of any of the variants.

To our knowledge, people do not have sufficient knowledge on UBA5 dimerization to speculate on whether some variants could play a dominant negative role. There is one variant, V260M, that lies at the dimer interface. We showed that the V260M variant biochemically affects ATP binding as well as UFM1 activation, but we do not have evidence to support that it causes dominant negative effects by affecting UBA5 dimerization.

Minor points:1. Page 5 line 45: It seems a reference is missing about the temperature dependence of Gal4 activity.

We apologize for the missing reference. We have incorporated a reference to PMID 25824290.

1. It might be of interest to assay the various transgenic rescue alleles at a higher temperature (say 29C) in addition to the nice work looking at 18/25C survival. Perhaps some of the alleles display temperature sensitivity at low (18) and high (29) temperatures.

We now include the survival rate data at 29C. The enzyme dead and severe LoF variants fail to rescue the lethality at 29C, while the mild (Group IA and IB) variants fully rescue. For the three Group II variants, the survival rate at 29C is higher than that at 25C and 18C. The results support the dosage sensitive effects of UBA5 overexpression, but do not support any variant to be temperature sensitive within this range.

**Reviewer #2**
Relative simplicity and genetic accessibility of the fly brain make it a premier model system for studying the function of genes linked to various diseases in humans. Here, Pan et al. show that human UBA5, whose mutations cause developmental and epileptic encephalopathy, can functionally replace the fly homolog Uba5. The authors then systematically express in flies the different versions of the gene carrying clinically relevant SNPs and perform extensive phenotypic characterization such as survival rate, developmental timing, lifespan, locomotor and seizure activity, as well as in vitro biochemical characterization (stability, ATP binding, UFM-1 activation) of the corresponding recombinant proteins. The biochemical effects are well predicted by (or at least consistent with) the location of affected amino acids in the previously described Uba5 protein structure. Most strikingly, the severity of biochemical defects appears to closely track the severity of phenotypic defects observed in vivo in flies. While the paper does not provide many novel insights into the function of Uba5, it convincingly establishes the fly nervous system as a powerful model for future mechanistic studies.One potential limitation is the design of the expression system in this work. Even though the authors state that "human cDNA is expressed under the control of the endogenous Uba5 enhancer and promoter", it is in fact the Gal4 gene that is expressed from the endogenous locus, meaning that the cDNA expression level would inevitably be amplified in comparison. The fact that different effects were observed when some experiments were performed at different temperatures (18 vs. 25) is also consistent with this. While I do not think this caveat weakens the conclusions of this paper, it may impact the interpretation of future experiments that use these tools, and thus should be clearly discussed in the paper. Especially considering the authors argue that most disease variants of UBA5 are partial loss-of-functions, the amplification effect could potentially mask the phenotypes of milder hypomorphic alleles. If the authors could also show that the T2A-Gal4 expression pattern in the brain matches well with that of endogenous RNA or protein (e.g. using HCR-FISH or antibody), it would help to alleviate this concern.

We thank the reviewer for pointing out the issue.

Regarding the humanization strategy we used in the study, we agree that this is a binary system which could induce overexpression of the target protein. However, as the reviewer also points out, this temperature sensitive system also enables us to flexibly adjust the expression level of the target protein (PMIDs 34113007, 35348658, 36206744), which is especially useful to study partial LoF variants. In our study we have successfully compared the relevant allelic strength of most of the variants.

We agree with the reviewer that a masking effect may exist in our system due to its gene overexpression nature. However, we cannot conclude that this masking effect really affects the three Group IA variants in our tests. The three variants are mild LoF, which is supported by our biochemical assays. Individuals homozygous for one of the Group IA variants, p.A371T, do not have any obvious phenotype, which is also consistent with our findings in flies.

Regarding the expression pattern of the T2A-GAL4, the Bellen lab has generated T2A-GAL4 lines for more than 3,000 genes. The expression pattern of many GAL4 lines faithfully reflect the expression pattern of the endogenous genes, which has been shown in our previous publications (PMIDs 25824290, 29565247, 31674908).

Recommendations For The Authors:As related to the expression pattern comment in the public review, I think the authors could also take advantage of Fly Cell Atlas or other available scRNA-seq atlases of the fly brain to present a much more detailed description of the Uba5 expression profile with minimal additional effort. If the cells that express it share other features or genes (other than the para that the authors mention), this could lead to further insights about the gene's neuronal or glial functions.

In response to the reviewer, we show the expression pattern of Uba5 documented in FlyCellAtlas and another adult brain single-cell RNA seq profile (PMID 29909982) in the revised manuscript.

In addition, one of the mutants (assuming the same one) is referred to as Leu254Pro in some parts of the manuscript while in some other parts (including tables 1-2) it is Lys254Pro.

We apologize for the mistakes. The variant should be Leu254Pro and we have made these corrections in the revised manuscript.

**Reviewer #3**
Summary:Variants in the UBA5 gene are associated with rare developmental and epileptic encephalopathy, DEE44. This research developed a system to assess in vivo and in vitro genotype-phenotype relationships between UBA5 allele series by humanized UBA5 fly models and biochemical activity assays. This study provides a basis for evaluating current and future individuals afflicted with this rare disease.Strengths:The authors developed a method to measure the enzymatic reaction activity of UBA5 mutants over time by applying the UbiReal method, which can monitor each reaction step of ubiquitination in real time using fluorescence polarization. They also classified fruit fly carrying humanized UBA5 variants into groups based on phenotype. They found a correlation between biochemical UBA5 activity and phenotype severity.Weaknesses:In the case of human DEE44, compound heterozygotes with both loss-of-function and hypomorphic forms (e.g., p.Ala371Thr, p.Asp389Gly, p.Asp389Tyr) may cause disease states. The presented models have failed to evaluate such cases.

We agree with the reviewer that our current system has a limitation that it evaluates one variant at a time rather than any combination of variants. However, our biochemical data do show that the three Group IA variants are mild LoF variants rather than benign variants. One of these variants, p.A371T, does not cause any obvious phenotype in homozygous individuals, which is also consistent with our findings in flies. The modeling of variant combinations, especially the Group IA/Group III combinations could be carried out in future studies.

Recommendations For The Authors:Figure 3G. Typo. "ContonS" should be replaced by "CantonS."

We apologize for the spelling mistake. We correct the typo in the revised manuscript.

Figure 5. The labels should be in uppercase instead of lowercase.

We correct the panel labels in the revised manuscript.

Figure 6A. Is the molecular weight of UBA5~UFM1 intermediate (99 kDa) in model Figure correct? In Fig. 6B, the molecular weight of UBA5~UFM1 intermediate seems to be 70-75 kDa.

Both are correct. The molecular weight depicted in the schematic of Figure 6A is based on the UBA5 dimer, which dissociates in the SDS-PAGE gel shown in Figure 6B. We have reconfigured the schematic to make this more apparent.

Figure. 6E, F, H, and I. The time points for quantification in these figures should be specified.

We apologize for the confusion. The details on data quantification are now more thoroughly explained in the Methods.